# Martian column $CO_2$ and pressure measurement with spaceborne differential absorption lidar at 1.96 µm

Zhaoyan Liu[1], Bing Lin[1], Joel Campbell[1], Jirong Yu[2], Jihong Geng[3], and Shibin Jiang[3]

[1]NASA Langley Research Center, Hampton, VA23681, USA
[2]Science Technology Corporation, Hampton, VA23666, USA
[3]AdValue Photonics, Inc, Tucson, AZ85714, USA

*Correspondence to*: Zhaoyan Liu (Zhaoyan.liu@nasa.gov)

**Abstract.** By utilizing progress in millijoule-level pulsed fiber lasers operating in the 1.96 µm spectral range, we introduce a concept utilizing a spaceborne differential absorption barometric lidar designed to operate within the 1.96 µm $CO_2$ absorption band for remote sensing of Martian atmospheric properties. Our focus is on the online wavelength situated in the trough region of two absorption lines, selected due to its insensitivity to laser frequency variations, thus mitigating the necessity for stringent laser frequency stability. Our investigation revolves around a compact lidar configuration, featuring reduced telescope dimensions and lower laser pulse energies. These adjustments are geared towards minimizing costs for potential forthcoming Mars missions. The core measurement objectives encompass the determination of column $CO_2$ absorption optical depth, columnar $CO_2$ abundance, surface atmospheric pressure, as well as vertical distributions of dust and cloud layers. Through the amalgamation of surface pressure data with atmospheric temperature insights garnered from sounders and utilizing the barometric formula, the prospect of deducing atmospheric pressure profiles becomes feasible. Simulation studies validate the viability of our approach. Notably, the precision of Martian surface pressure measurements is projected to surpass 1 Pa when the aerial dust optical depth is projected to be under 0.7, a typical airborne dust scenario on Mars, considering a horizontal averaging span of 10 km.

## 1 Introduction

Atmospheric temperature and pressure play pivotal roles in determining the states and dynamics of extraterrestrial planetary atmospheres within the solar system. The temperature structure of the atmosphere is governed by dynamics, particularly the heating from solar radiation and thermal emission from the surface and atmosphere. Meanwhile, pressure and pressure gradients serve as the primary driving forces for atmospheric motion and the transport of mass, heat, and momentum (Holton, 1979). Air movements on extraterrestrial planets represent crucial atmospheric dynamic processes, heavily influenced by radiative heating and pressure fields. In the case of the Martian atmosphere, these dynamic processes can produce synoptic scale storm systems characterized by significant winds and dusts activity. Accurate observation, modeling, and prediction of

temperature, pressure, and dust aerosol fields on Mars are vital for understanding Martian weather systems, particularly the occurrence of dust storms. Moreover, these efforts provide invaluable support for safe and accurate atmospheric entry, suitable landing site selection, and ultimately human colonization of Mars. The Martian atmosphere is predominantly composed of carbon dioxide (approximately 95.1%), along with small amounts of nitrogen and argon (approximately 2.59% and 1.94%, respectively). It also contains traces of oxygen, water vapor, carbon monoxide, and other noble gases (Williams, 2020). Understanding the global atmospheric dynamics, dust storms, and variations in the carbon cycle on Mars is crucial for successful Mars exploration (Spiga et al., 2018). Remote sensing techniques may be the only practical means to observe and gain knowledge about these processes and variations.

The global measurement of planetary atmospheric temperature can be achieved through infrared (IR) remote sensing techniques. Martian atmospheric temperature soundings have a historical precedence, exemplified by instruments such as the spaceborne TES (Thermal Emission Spectrometer). This instrument effectively captured infrared emissions from Mars, proving well-suited for retrieving the atmospheric thermal structure (Smith et al., 2001). For the Martian atmosphere, surface pressure measurements have been conducted at specific locations using in-situ barometers on missions such as Viking Landers, Mars Pathfinder, Phoenix Mars Lander, Mars Science Laboratory, and the recent InSight mission (Banfield et al., 2020). Passive remote sensing instruments like the near-infrared imaging spectrometer (Forget et al., 2007; Spiga et al., 2007), Mars Express OMEGA visible and near-infrared imaging spectrometer (Forget et al., 2007; Spiga et al., 2007), and the IMS (Imaging Spectrometer) abord Phobos2 (Bibring et al., 1991) made it possible to do large-scale surface pressure mapping on Mars and have been used to measure the amount of $CO_2$ in the 2-μm CO2 absorption band using reflected solar radiation.

While surface pressure observations obtained through passive remote sensing techniques offer valuable insights into the dynamics of the Martian atmosphere, they have certain limitations. Critical issues of these techniques include: (1) Atmospheric pressure measurements using the passive technique can only be performed during daytime, restricting the temporal coverage of observations. (2) The absence of ranging capability in passive measurements may introduce systematic errors. Dust and cloud reflections can result in different path lengths of sunlight compared to surface reflections, leading to uncertainties in the derived pressure values. (3) Observations of $CO_2$ changes and pressure fields are unavailable in certain crucial regions, such as the two polar regions. This limitation hinders our understanding of the Martian carbon cycle, dynamics, and the interaction between polar regions and lower latitudes. (4) Passive measurements are confined to surface pressure fields and cannot account for sufficient information regarding surface elevation variations. The Martian atmosphere is characterized by ubiquitous airborne dust, which can interfere with passive measurements. Additionally, the terrain surface on Mars exhibits significant changes at various spatial scales (Frey et al., 1998; Smith et al., 1999), potentially introducing bias into passive measurements. Our previous study (Lin and Liu, 2021) proposed the integration of active sensors into the existing suite of pressure sensing instruments for Martian atmospheric studies and Mars exploration. Specifically, a pulsed $CO_2$ differential absorption lidar (DIAL) operating at the 2.05-μm $CO_2$ absorption band was simulated and evaluated. Unlike passive sensors that are limited to column $CO_2$ measurements, which can be biased by the presence of clouds and/or dust aerosols, a pulsed DIAL system enables the collection of range-resolved return signals from all atmospheric backscattering targets, including aerosols, clouds, and the

surface. Consequently, when combined with a thermal infrared temperature sounder, the DIAL system has the potential to provide vertical profile measurements of pressure (Lin and Liu, 2021). Furthermore, a DIAL system offers advantages such as suitability for pressure measurements over varying topography, the ability to operate during both day and night, and the capability to obtain measurements over polar regions. It is noteworthy that deploying active optical remote sensing instruments for space measurements typically incurs higher costs compared to passive ones. Therefore, adopting a strategy that involves utilizing active remote sensing instruments with support from lightweight, compact, and low-cost passive remote sensing instruments would be advantageous. This approach is exemplified in the CALIPSO mission, where the payload includes one active backscatter lidar, infrared imaging radiometer, and a wide-field camera (Hunt et al., 2009).

This study reexamines the concept of Martian pressure measurement using a $CO_2$ DIAL on an orbiter and explores additional opportunities within the 1.96 µm $CO_2$ absorption band. Recent advancements in all-fiber lasers have demonstrated the generation of millijoule-level pulses with kilohertz repetition frequencies, enabling high transmitted powers at this wavelength band.

## 2. DIAL System and Methodology

### 2.1 DIAL Measurement

In the $CO_2$ differential absorption measurement, it is common practice to choose two or more wavelengths (Abshire et al., 2010; Lin et al., 2013; Refaat et al., 2015). One wavelength, referred to as the offline wavelength, is selected far from the center of the absorption line where the $CO_2$ absorption is insignificant, serving as a baseline reference. The other wavelengths, known as the online wavelengths, are chosen on the line where the $CO_2$ absorption is substantial, enabling accurate measurement of $CO_2$ concentrations. The selected online and offline wavelengths are closely positioned, ensuring that the differences in optical depths due to attenuations other than $CO_2$ absorption (mainly scattering optical depths) between these wavelengths are negligibly small. Consequently, the one-way $CO_2$ differential absorption optical depth (DAOD) can then be determined by calculating the ratio of received signals at the online and offline wavelengths (Lin and Liu, 2021):

$$\Delta\tau_{CO_2} = -\frac{1}{2}ln\left(\frac{N_{s,on}}{N_{s,off}}\frac{C_{0,off}}{C_{0,on}}\right). \tag{1}$$

$N_{s,\text{on}}$ and $N_{s,\text{off}}$ correspond to the received signal photons within a sample gate $\Delta t$ at the online and offline wavelengths, respectively. $C_{0,\text{on}}$ and $C_{0,\text{off}}$ denote the system constant which contains lidar system parameters and other range-independent quantities (refer to Eq. (A3)), for the online and offline wavelengths, respectively. In practical applications, the ratio of $C_{0,\text{off}}$ to $C_{0,\text{on}}$ is required to derive $\Delta\tau_{CO2}$. This ratio can be determined by the ratio of online and offline signals backscattered from a target close to the lidar or a target where the $CO_2$ absorption is effectively zero (Lin et al., 2015; Dobler et al., 2013; Campbell et al., 2020). For the spaceborne DIAL measurement, the $CO_2$ DAOD can be expressed as:

$$\Delta\tau_{CO_2} = \int_z^{TOA}\left[\alpha_{CO_2,on}(z') - \alpha_{CO_2,off}(z')\right]n_{CO_2}(z')dz' = A_{CO_2}(z)N_{CO_2}(z), \tag{2}$$

where $\alpha_{CO2}(z)$ and $n_{CO2}(z)$ are the $CO_2$ absorption cross section and number density at altitude $z$, respectively. $N_{CO2}(z)$ is the column $CO_2$ molecular number integrated from $z$ to the top of atmosphere (TOA) which refers to the outer boundary of Mars's atmosphere. In the simulation for this study, the upper limit of the integral is set at 60 km, where the pressure is approximately 0.14% of that at the surface. $A_{CO2}(z)$ is a mean differential absorption cross section from $z$ to TOA weighted by $CO_2$ number density,

$$A_{CO_2}(z) = \frac{\int_z^{TOA} \left[ \alpha_{CO_2,on}(z') - \alpha_{CO_2,off}(z') \right] n_{CO_2}(z') dz'}{\int_z^{TOA} n_{CO_2}(z') dz'}. \tag{3}$$

From Eq. (2), we derive:

$$N_{CO_2}(z) = \frac{\Delta\tau_{CO_2}(z)}{A_{CO_2}(z)}, \tag{4}$$

and this column molecular number density, $N_{CO_2}(z) = \int_z^{TOA} n_{CO_2}(z') dz'$, determines the $CO_2$ partial atmospheric pressure at z:

$$P_{CO_2}(z) = M_{CO_2} g_W(z) N_{CO_2}(z) \tag{5}$$

where $M_{CO2}$ is the $CO_2$ molecular mass, and

$$g_W(z) = \frac{\int_z^{TOA} g(z') n_{CO_2,model}(z') dz'}{\int_z^{TOA} n_{CO_2,model}(z') dz'} \tag{6}$$

is the mean Martian gravitational acceleration between $z$ and TOA, $n_{CO2}(z)$ is the $CO_2$ number density and $g(z)$ the gravitational acceleration at altitude $z$. The Martian atmospheric pressure is the sum of $CO_2$ pressure and the pressure of all other gases $P_{other}(z)$

$$P(z) = P_{CO_2}(z) + P_{others}(z). \tag{7}$$

The Martian atmosphere is predominantly composed of carbon dioxide. The pressure exerted by other gases on Mars, $P_{others}$, is small and remains relatively stable (Trainer et al., 2019) compared to the significant deposition and sublimation activities of $CO_2$. The determination of $P_{others}$ can be achieved through climatological analysis, modeling, or other available measurements. In our previous study (Lin and Liu, 2021), we proposed a DIAL system with a telescope size of 1 m and a laser pulse energy of 5 mJ at the online wavelengths. The system was designed for atmospheric profiling and column measurements of $CO_2$ and pressure. However, in this current study, our focus is on the measurement of the crucial dynamic variable, surface atmospheric pressure. Consequently, we can reduce the telescope size and laser pulse energy to build a more compact lidar system. As illustrated by the simulation results presented in Section 3.2, the reduction in telescope size and pulse energy presents challenges for accurately measuring atmospheric $CO_2$ DAOD. However, the surface return signal remains sufficiently strong

to enable precise measurement of $CO_2$ DAOD, from which surface pressure can be accurately derived. The surface return signal is consistently available when the airborne dust load is low or moderate.

**2.2 Atmospheric Pressure Measurement with IR Sounder Temperature Measurement**

While the DIAL system considered in this study may not allow for atmospheric vertical profiling of $CO_2$ DAOD due to weaker lidar return signals compared to our previous study (Lin and Liu, 2021, and Section 3), it is still possible to derive vertical
profiles of atmospheric pressure. This can be accomplished by leveraging DIAL surface pressure measurements and thermal infrared (IR) temperature profile measurements. However, it is important to note that a simultaneous temperature profile measurement is not mandatory in DIAL surface pressure measurements, although it can enhance measurement accuracy. Thermal IR sounders are compact and cost-effective sensors commonly used for satellite temperature measurements (Kalmus et al., 2022; Natraj et al., 2022).
Given the temperature $T(z)$ measured by the sounder at a specific altitude z, the pressure can be determined using the barometric formula:

$$P(z) = P_0 e^{-\int_0^z \frac{M_{Mars} g(z)}{RT(z)} dz} . \tag{8}$$

Where $P_0$ represents a reference pressure at the surface or an altitude where a $CO_2$ DAOD is available from a dense dust layer or cloud, $M_{Mars} = 0.04334$ kg/mole denotes the molar mass of Martian atmosphere (Williams et at., 2020), and $R = 8.314$ JK$^{-1}$
$^1$mol$^{-1}$ represents the universal gas constant.

To calculate $P(z)$ using Eq. (8), the reference pressure $P_0$ is required. $P_0$ can be obtained from the DIAL $CO_2$ DAOD measurement at the surface. However, to accurately calculate the weighting function $A_{CO2}$ in Eq. (3) and, subsequently, $P_{CO2}$ in Eq. (5), some knowledge about the temperature and pressure profiles is necessary. Hence, an iterative procedure is applied. Initially, a climatological or modeled value $P_{0c}$ can be used as an estimate for the surface pressure to calculate the first set of
$P(z)$ using Eq. (8). With the initially calculated $P(z)$ and the temperature profile $T(z)$ obtained from the sounder measurements, the weighting function $A_{CO2}(z)$ in Eq. (3) and, consequently, the $CO_2$ pressure at the surface $P_{0,CO2}$ can be retrieved more accurately from the $CO_2$ DAOD measurement using Eq. (5). As previously mentioned, $CO_2$ comprises the dominant composition of the Martian atmosphere, and the pressure of other gases ($P_{others}$) remains relatively stable. Using $P_{0,CO2}$, the surface pressure $P_0$ can then be determined from $P_{0,CO2}$ using Eq. (7). Once $P_0$ is determined from the $CO_2$ DAOD
measurement, it can replace the climatological or modeled value $P_{0,c}$ to recalculate $P(z)$ using Eq. (8). This process can be repeated iteratively to improve the retrieval of $P_0$. In cases where very dense dust or cloud layers are present, the surface return may not be available. However, accurate $CO_2$ DAOD measurements may be derived from the lidar return signals from these targets. The pressure $P$ at altitude $z_0$, where the dense dust or cloud layers are located, can be used as the reference $P_0$ in Eq. (8). The afore mentioned iterative procedure can then be applied to derive the pressure $P$ at $z_0$, above and below.

## 2.3 Wavelength Selection

Considering the size, weight, and power (SWaP) constraints crucial for space-based systems, particularly for Mars mission lidars, this study utilizes a compact telescope and focuses on the column $CO_2$ and pressure measurement. Furthermore, the online wavelength is selected at the trough region between two adjacent absorption lines. This choice can alleviate sensitivity to laser frequency variability (Korb and Weng, 1983), thereby easing the stringent requirements for laser frequency stability and reducing costs. The wavelengths selected in the 2.05 μm absorption band, specifically line R(32) of the ν'(20013) vibrational band with the line center at 2.050428 μm, in our previous study (Lin and Liu, 2021) are very close to those (line R(30) of the same vibrational band) used in NASA Langley Research Center's (LaRC's) pulsed DIAL systems for $CO_2$ measurement on Earth (Refaat et al., 2015; Yu et al., 2017). However, the differences in line selection reflect the distinct atmospheric environments of Earth and Mars. Compared to Earth's atmosphere with a carbon dioxide ($CO_2$) volume mixing ratio of approximately 400 parts per million (ppm), the Martian atmosphere is predominantly composed of $CO_2$, accounting for about 95.3% by volume, despite having a much lower total amount or atmospheric pressure. As a result, pressure-induced absorption line broadening is significantly smaller in the Martian atmosphere, and the line shape is much narrower. Consequently, the lidar wavelengths are typically chosen on the wing of an absorption line where the column $CO_2$ DAOD falls within the range of 0.5-2, with an optimal DAOD value of 1.11 (Lin and Liu, 2021).

The criteria for wavelength selection in this study are as follows: (1) the presence of a strong absorption line with a nearby weak absorption line on the wing of the strong line, (2) both the strong and weak lines are from the principal isotope $^{12}C^{16}O_2$, (3) AOD of the trough region is approximately 1.1, and (4) there is no absorption from other gases. While weak absorption lines from $CO_2$ isotopes other than $^{12}C^{16}O_2$ could also be selected, the accuracy of current knowledge regarding the abundance of these isotopes on Mars may impact spectroscopic analysis. In this study, the strong absorption line selected is P(10) from the $^{12}C^{16}O_2$ vibration band (ν' = 20011, with a center wavelength of 1.9640146 μm), as shown in Fig. 1. The $CO_2$ absorption is calculated using a typical Martian atmosphere obtained from Viking 1 observations (Seiff and Kirk, 1977), as illustrated in Fig. 2. The online wavelength is set at the trough region near 1.9639572 μm of the selected strong line, along with a weak line centered at 1.9639502 μm from another $^{12}C^{16}O_2$ vibration band, as shown in Fig. 3. The modeling of the Martian atmosphere in Fig. 3 is based on the Viking 1 observation depicted in Fig. 2. The AOD is calculated using the line-by-line calculation method through the HITRAN Application Programming Interface (HAPI) (Kochanov et al., 2016). All absorption lines of all $CO_2$ isotopes within the wavenumber range of 5063 - 5128 cm$^{-1}$, covering the entire absorption band, are taken into account in the HAPI line-by-line calculation. The calculation is performed with a wavelength resolution of 5x10$^{-5}$ cm$^{-1}$ (equivalent to ~2x10$^{-5}$ nm), utilizing the Voigt distribution with a wing length of 10 cm$^{-1}$. Figure 3b, essentially a derivative of Fig. 3a, illustrates that the sensitivity of AOD to laser frequency variability is significantly smaller in the trough regions compared to the surrounding regions. This is because the derivative or slope at the trough or peak region is close to zero. Thus, in this study, the online wavelength is set in the trough region of the two selected lines. The column $CO_2$ AOD at the trough is approximately 0.885, corresponding to a DAOD of 0.825, which is close to the optimal value of 1.11. The change in online AOD is smaller

than $10^{-4}$ for a 1 MHz change in laser frequency at the selected trough region (refer to Fig. 3b), and the change in offline AOD is even smaller. These small AOD changes due to laser frequency variations lead to insignificant errors in DAOD calculations and retrievals. Consequently, the requirements for laser frequency stability can be relaxed.

It is worth noting that P(12) of the same 12C16O2 ν'(20011) vibration band could also be considered a good candidate, as it has a few weak absorption lines on its wing. However, in this study, our analysis focuses solely on P(10).

### 2.4 Laser and Wavelength Locking

To attain high measurement precision, it is essential to stabilize the laser frequency, as discussed in numerous papers (e.g., Refaat et al., 2015). The NASA LaRC has made significant advancements in laser frequency control and locking techniques over the past decades. Figure 4 illustrates a conceptual diagram depicting the master laser wavelength locking and control system. For this system, two or more continuous-wave single-frequency fiber or semiconductor-distributed feedback lasers can be utilized as master lasers. In this setup, one master laser is locked at the center of the selected line at 1.9640146 μm. The Pound-Drever-Hall frequency stabilization scheme is employed with a $CO_2$ absorption cell to achieve this locking. The other master laser, or possibly two master lasers, can be locked off the line center by 4.44 GHz for the online wavelength or -25.75 GHz for the offline wavelength. The seed lasers employed in the LaRC airborne $CO_2$ DIAL system can be wavelength-locked at the line center or locked up to 35 GHz from the line center, with a long-term frequency jittering of 0.3 MHz (Refaat et al., 2015; Koch et al., 2008). With the laser frequency stability achieved at this level and the online laser wavelength in the trough region, the error in DAOD due to laser frequency variability is smaller than $10^{-4}$ for an 80 MHz range, making it insignificant for DIAL DAOD measurements.

The laser transmitter considered in this study is an all-fiber master oscillator power amplifier (MOPA) system. Optical fiber amplifier technology has advanced considerably. Specifically, pulsed laser energy exceeding 1 mJ at kilohertz repetition frequencies has been demonstrated in the 1.97 μm band. These technological breakthroughs enable the development of compact and lightweight laser sources for future Mars missions.

## 3 Simulation

### 3.1 Error Analysis

Based on the first order error propagation theory, the error $\varepsilon$ and the relative error due to random noise for all individual quantities in Eqs. (3) – (5) can be estimated (Lin and Liu, 2021) using:

$$\varepsilon_{\Delta\tau_{CO_2}(z)} = \frac{1}{2}\left(\left(\frac{1}{SNR_{on}(z)}\right)^2 + \left(\frac{1}{SNR_{off}(z)}\right)^2\right)^{1/2},$$  (9a)

$$210 \quad \frac{\varepsilon_{\Delta\tau_{CO_2}(z)}}{\Delta\tau_{CO_2}(z)} = \frac{1}{2\Delta\tau_{CO_2}(z)}\left(\left(\frac{1}{SNR_{on}(z)}\right)^2 + \left(\frac{1}{SNR_{off}(z)}\right)^2\right)^{1/2}, \tag{9b}$$

$$\varepsilon_{N_{CO_2}(z)} = \sqrt{\left(\frac{\sigma_{\Delta\tau_{CO_2}(z)}}{A_{CO_2}(z)}\right)^2 + \left(N_{CO_2}(z)\frac{\sigma_{A_{CO_2}(z)}}{A_{CO_2}(z)}\right)^2}, \tag{10a}$$

$$\frac{\varepsilon_{N_{CO_2}(z)}}{N_{CO_2}(z)} = \sqrt{\left(\frac{\varepsilon_{\Delta\tau_{CO_2}(z)}}{\Delta\tau_{CO_2}(z)}\right)^2 + \left(\frac{\varepsilon_{A_{CO_2}(z)}}{A_{CO_2}(z)}\right)^2}, \tag{10b}$$

$$\varepsilon_{P_{CO_2}(z)} = M_{CO_2} g_W(z) \varepsilon_{N_{CO_2}(z)}, \tag{11a}$$

$$\frac{\varepsilon_{P_{CO_2}(z)}}{P_{CO_2}(z)} = \frac{\varepsilon_{N_{CO_2}(z)}}{N_{CO_2}(z)} = \frac{\varepsilon_{\Delta\tau_{CO_2}(z)}}{\Delta\tau_{CO_2}(z)}. \tag{11b}$$

Where $\varepsilon_x$ represents the uncertainty or error for parameter x. Equation (11b) demonstrates that the relative errors in the measured $CO_2$ DAOD are equivalent to the corresponding relative errors in the observations of $CO_2$ amount and atmospheric pressure. As discussed in Section 2.3, the vertical profile of atmospheric pressure can be derived using Eq. (8) when the atmospheric temperature profile is measured using an IR sounder. The error and relative error in the retrieval of atmospheric pressure can be estimated using the following equations:

$$220 \quad \varepsilon_{P(z)} = \sqrt{\left(e^{-\int_0^z \frac{M_{Mars}g(z)}{RT(z)}dz} \cdot \varepsilon_{P_0}\right)^2 + \left(P_0 e^{-\int_0^z \frac{M_{Mars}g(z)}{RT(z)}dz}\int_0^z \frac{M_{Mars}g(z)}{RT^2(z)}dz \cdot \varepsilon_{T(z)}\right)^2} \tag{12a}$$

$$\frac{\varepsilon_{P(z)}}{P(z)} = \sqrt{\left(\frac{\varepsilon_{P_0}(z)}{P_0}\right)^2 + \left(\int_0^z \frac{M_{Mars}g(z)}{RT(z)}\frac{\varepsilon_{T(z)}}{T(z)}dz\right)^2}. \tag{12b}$$

The first term in Eq. (12a) and (12b) represents the contribution of the error in the surface pressure $P_0$, which is used to calculate the atmospheric pressure $P$ using Equation (8). This term includes the error in the retrieved $P_{0,CO2}$ from the $CO_2$ DAOD measurement and the error in the pressure of other gases ($P_{others}$). While it is challenging to estimate the exact error in $P_{others}$, it
is anticipated to be small. This is because $P_{others}$ is relatively stable and constitutes only a small fraction of the total Martian atmospheric pressure ($< 5\%$) (Williams et al., 2020).

Random errors in the measured temperature $T$ can be partially smoothed out through the integration calculation in Eq. 8. However, it's important to note that any systematic error in $T$ cannot be reduced by integration or signal averaging, and such systematic errors can propagate into the retrieval of atmospheric pressure $P$ using Eq. 8. A study by Natraj et al. (2022)
demonstrated that the total error in temperature measurements for the JPL GEO-IR Sounder ranges from 0.3-1 K, with a

precision of 0.1-0.3 K. Data fusion techniques that combine measurements from multiple satellite sounders have been shown to reduce bias in near-surface temperature measurements, resulting in mean biases smaller than 0.16 K (Kalmus et al., 2022). Considering these low bias errors, this study conservatively assumes potential temperature bias errors within 2 K.

The error and relative error in atmospheric pressure $P$ (i.e., the second term in Eq. (12a) and (12b)) can be estimated as a function of altitude, considering different biases in temperature $T$ of $\pm0.5$ K, $\pm1.0$ K, and $\pm2.0$ K. These estimations are presented in Fig. 5. From Fig. 5, it can be observed that the bias in temperature $T$ has a minimal impact on the retrieval of atmospheric pressure near the surface. However, as altitude increases, the influence of the temperature bias on pressure retrieval becomes relatively more significant (Fig. 5a). This is due to the cumulative effect of the temperature bias as altitude increases (via the integration term in Eq. 12). The absolute error in atmospheric pressure $P$ (as shown in Fig. 5b) due to the temperature bias is very small near the surface. It gradually increases with altitude, reaching a maximum around 12 km, and then decreases. This trend is primarily driven by the decreasing trend of atmospheric pressure with increasing altitude.

Systematic errors or biases in temperature can further propagate to the retrieval of $CO_2$ pressure ($P_{CO2}$) using Eq. (5) through the calculation of the weighting function ($A_{CO2}$). To assess the impact of $T$ biases on $P_{CO2}$, errors and relative errors in $P_{CO2}$ as a function of altitude are simulated using the HAPI software for $T$ biases of $\pm0.5$ K, $\pm1.0$ K, and $\pm2.0$ K. The results are presented in Fig. 6. In Fig. 6, the curves represent the systematic errors or relative systematic errors in $P_{CO2}$ at different altitudes resulting from the $T$ bias. The magnitudes of the relative errors initially decrease with increasing altitude until approximately 1.4 km, after which they start to increase. At z = 0 (representing the surface in this study), the magnitudes of the relative errors in $P_{CO2}$ are smaller than 0.1% (Fig. 6a), and the absolute errors are smaller than 0.5 Pa (Fig. 6b) when the $T$ bias is smaller than 2 K. The magnitudes of errors reach a maximum around 14 km, remaining below 1.6 Pa when the $T$ bias is smaller than 2 K.

## 3.2 Simulation Results

When comparing the $CO_2$ DIAL measurement on Mars to the CALIPSO lidar measurement on Earth (Hunt et al., 2009), several advantages of space lidar measurements on Mars can be observed. These advantages are summarized in Table 1 and quantified by a figure of merit (FOM) in terms of SNR improvement. On Mars, the smaller size and mass of the planet, as well as the greater distance from the Sun, contribute to the advantages of space lidar measurements. A lower orbit height on Mars results in a FOM of 8.05, as the received signal is proportional to the squared range from the lidar to the atmospheric backscatter. Slower ground speed on Mars allows for a longer averaging time for a given horizontal distance, leading to a FOM of 2.17. Moreover, the photon number per unit pulse energy is 3.9 times larger at 1.964 μm compared to the visible region at 0.532 μm. The combined effect of these factors yields a FOM of approximately 67, which is a significant advantage. This allows for the use of lower laser power and/or smaller receiving telescope size, resulting in a more compact lidar system. Furthermore, the solar radiation constant on Mars is 2.3 times smaller than on Earth, and the solar radiation at 1.964 μm is approximately 220 times smaller than in the visible region. As a result, the daytime background noise, which is the dominant noise source in

visible lidar measurements on Earth, is significantly reduced in the Mars measurement. This reduction in noise further enhances the performance of the DIAL system.

Considering these advantages, a compact DIAL system for Mars can be developed with relatively small telescope size and laser pulse energy. The specific parameters are listed in Table 2, and they can be achieved using currently available technologies, parts, and devices.

To assess the impact of detection noises and evaluate the performance of the DIAL system, observing system simulation experiments (OSSEs) are conducted. These OSSEs are based on the system parameters listed in Table 2. It should be noted

that these parameters are similar to those used in Lin and Liu (2021), with a few differences. In this study, the telescope size is reduced to 0.3 meters, which is approximately 3.3 times smaller than in the previous study. Additionally, the laser output energy for both the online and offline wavelengths is set to 1.5 mJ, whereas the previous study assumed values of 5 mJ for the online wavelength and 2 mJ for the offline wavelength. It is worth mentioning that recent advancements in all-fiber MOPA lasers have demonstrated laser output energies close to 2 mJ at the selected wavelengths, supporting the feasibility of the

system parameters proposed in this study.

The OSSEs specifically focus on random errors caused by detection noises. The experiments provide valuable insights into the system's sensitivity, accuracy, and overall performance, thereby guiding further improvements and developments in the field of DIAL technology for Mars exploration.

For the detector in the DIAL system, a HgCdTe Avalanche Photodiode (APD) manufactured by Leonardo DRS Electro-Optical

Infrared Systems, referred to as DRS APD hereafter, is assumed, which is currently used for spaceborne lidar applications in the IR region (Lin et al., 2013; Sun et al., 2017). The selection of this detector is based on its suitability for the desired performance and requirements of the lidar system. Furthermore, the optical parameters of the lidar system, including the field of view (FOV), beam expander, and transceiver throughputs, are adapted from the CALIPSO backscatter lidar. CALIPSO is a backscatter lidar that was launched in 2006 (Hunt et al., 2009) and has been nearly continuously operating in space. By utilizing

these established optical parameters, the DIAL system can benefit from the experience and success of the CALIPSO mission. These optical parameters, along with the choice of a suitable detector, contribute to the overall design and performance of the DIAL system, enabling accurate and reliable measurements of atmospheric parameters for Mars exploration.

The OSSE results are presented in Fig. 7. Figure 7a shows the modeled dust profile based on the dust vertical distribution observed by the SPICAM occultation measurement on Mars-Express (Fedorova et al., 2009). The occultation measurement

was conducted above 10 km, and the dust distribution was extrapolated to the surface using an exponential curve. The modeled dust column optical depth from TOA to the surface is 0.373 at 1.964 μm. In the presence of the modeled dust, the online signal-to-noise ratio (SNR) is too small in the atmosphere to accurately profile $CO_2$ DAOD and pressure, as shown in Fig. 7b. However, during nighttime, the offline SNR is greater than 10 below ~14 km with a horizontal resolution of 10 km and vertical resolution of 1 km. With further averaging, if needed, these measurements can provide dust profiles in the lower atmosphere.

For the daytime simulation, the online and offline SNRs experience a decrease due to the presence of solar background noise. However, the decrease is not significant compared to the nighttime SNRs. This is because the solar background radiation at

1.964 µm is significantly smaller compared to the CALIPSO aerosol measurement at 532 nm (Hunt et al., 2009). It is important to note that the worst-case scenario of a Sun zenith angle (SZA) of 30°, corresponding to the SZA at the equator where the background noise is the strongest, is considered in the daytime simulation. This indicates that the DIAL system can maintain reasonable SNRs even in the presence of solar background noise during daytime operations. These OSSE results provide valuable insights into the performance and limitations of the DIAL system in the presence of dust and under different lighting conditions.

Dust on Mars exhibits significant annual and geophysical variations, leading to fluctuations in dust optical depth (OD) ranging from approximately 0.4 to 1.4 at 880 nm (Chen-Chen et al., 2019). During the Mars year 34 global dust storm, the dust OD was as high as approximately 8 (Guzewich et al., 2019). In cases of heavy dust loading, the DIAL system considered in this study may enable measurements of $CO_2$ DAOD and pressure in the Martian atmosphere. While the SNR in the atmosphere is generally too low to achieve accurate measurements of Martian $CO_2$ DAOD and pressure, the lidar return signal from the surface is several orders of magnitude stronger. This allows for precise retrieval of column $CO_2$ DAOD and surface atmospheric pressure from the lidar surface return. Figures 7c and 7d present lidar return signals from the surface and surface return SNR, respectively, simulated using Eq. (A2b) and Eq. (A1) for a single laser shot. Due to the strong CO2 absorption for the online wavelength, the return signal is smaller at the online wavelength than at the offline wavelength, resulting in a smaller SNR at the online wavelength than at the offline wavelength. However, the SNR is similar during the day and night due to the dominance of lidar signal noise for both online and offline signals. Figure 7e and 7f illustrate the relative error and error, respectively, in surface $CO_2$ DAOD ($\Delta\tau_{CO2}$) and pressure $P_{CO2}$ simulated for both nighttime and daytime scenarios at a horizontal resolution of 10 km using Eq. (9) and Eq. (11).

Interestingly, the curves for nighttime and daytime measurements closely overlap due to the strong surface return signal, which results in signal shot noise dominating the measurement when a commercial solar blocking filter of 0.8 nm is utilized. The relative error in $CO_2$ DAOD remains below 0.2% when the dust OD is approximately 1 or lower. Similarly, the error in $P_{0,CO2}$ stays below 1 Pa when the dust OD is around 0.7 or lower. As the dust OD increases, the error in both measurements rises, but remains below 1.6 Pa until the dust OD reaches 1.

## 4 Conclusions

In our previous study, we proposed a novel concept utilizing a differential absorption barometric lidar operating at the 2.05 µm $CO_2$ absorption band ($v' = 20013$) for remote sensing of Martian atmospheric $CO_2$ amount and atmospheric pressure (Lin and Liu, 2021). The present study expanses the selection of laser wavelengths to the 1.96 µm $CO_2$ absorption band ($v' = 20011$) to leverage the recent advancements in millijoule-level pulsed fiber lasers at this wavelength. Furthermore, the online wavelength is set at the trough region of two absorption lines, where the $CO_2$ AOD exhibits insensitivity to the laser frequency variability. This characteristic significantly relaxes the requirement for laser frequency stability. Our measurements will focus on column $CO_2$ differential aerosol optical depth (DAOD), column CO2 amount, and surface pressure using a compact

telescope with a size of 0.3 m and a laser pulse energy of 1.5 mJ at a repetition frequency of 2 kHz. With this considered differential absorption lidar (DIAL) system, we can retrieve $CO_2$ pressure at the surface or in the atmosphere where dense dust/cloud layers are present during both day and night. Additionally, the atmospheric pressure profiles can be derived by combining the DIAL surface pressure measurements with atmospheric temperature observations obtained from sounders based on the barometric formula. Furthermore, the observation of dust and cloud vertical distributions at low altitudes is possible. OSSE simulations were performed to estimate noise-induced random error. The results indicate that a relative error smaller than 0.2% is achievable for surface $CO_2$ DAOD and pressure $P_{CO2}$ measurements at a horizontal average of 10 km when the airborne dust OD is small than 1, a condition in which the Martian airborne dust is commonly observed. An error for $P_{CO2}$ smaller than 1 Pa is possible at the surface when the dust OD is smaller than 0.7. Achieving such measurement precision would facilitate the collection of crucial data for Mars's climate studies, enabling the acquisition of dynamic information to enhance forecasts of Martian weather and climate systems. Furthermore, future efforts in instrumentation development and exploration of atmospheric $CO_2$ measurements would expand the application to Martian atmospheric entry, landing site selection, severe dust storm prediction, and ultimately future human missions.

Appendix A Signal-to-Noise Ratio Calculation

The Signal-to-Noise Ratio (SNR) for a lidar using analogue detection with an Avalanche Photodiode (APD) detector can be calculated according to [Liu et al., 2000]:

$$SNR(r) = \frac{N_s(r)}{\sqrt{(N_s(r)+N_b+N_d)F_m+\frac{I_c^2\Delta t}{M^2 q^2}}}. \tag{A1}$$

Where $N_s(r)$ is the received signal in photon counts at a range $r$ from the lidar, and is given by [Lin and Liu, 2021]:

$$N_s(r) = \frac{1}{r^2}\frac{c\Delta t}{2}C_0\,\beta(r)exp\big[-2\big(\tau_{CO2}(r)+\tau_{other}(r)\big)\big], \tag{A2a}$$

for the atmospheric return, and

$$N_s = \frac{1}{r^2}C_0\,R_{surf}exp\big[-2\big(\tau_{CO2}(r)+\tau_{other}(r)\big)\big], \tag{A2b}$$

for the surface return, where $\beta$ and $R_{surf}$ are the total backscatter in the atmosphere and the surface reflectance, respectively, c the speed of light, $\Delta t$ the sampling time gate, and $\tau_{CO2}$ and $\tau_{other}$ are, respectively, the one-way optical depth due to the $CO_2$ absorption and the other attenuation due primarily to atmospheric scattering. $C_0$ is the lidar system constant containing all the system parameters and other range-independent quantities,

$$C_0 = E_0 A_0 k_0 \eta_Q^* \frac{\lambda}{hc} \tag{A3}$$

where $E_0$ is the laser energy per pulse, $A_0$ the telescope area, $k_0$ the system optical throughput, $\eta_Q^*$ the effective detector quantum efficiency (a product of the quantum efficiency and fill factor in Table 2 for the DRS APD), $\lambda$ the laser wavelength, and $h$ the Planck's constant.

$N_b$ in Eq. (A1) is the received solar background in photon counts and is given by

$$N_b = I_b A_0 \Delta\lambda \Delta t k_0 \eta_Q^* \frac{\phi^2 \pi}{4} \frac{\lambda}{hc},$$  (A4)

where $I_b$ is the solar spectral radiance reflected from the atmosphere and the surface, $\Delta\lambda$ is the bandwidth of the solar blocking filter and $\phi$ is the receiver's field of view.

$N_d$ is the dark count of the APD detector calculated using

$$N_d = \frac{I_d \Delta t}{q}$$  (A5)

where $I_d$, $M$ and $F_m$ are the unity-gain dark current, gain and the excess noise factor of the APD detector, respectively, $q$ is the electron charge, and $I_c$ is the spectral noise current density of electronic circuit which can be ignored when $M$ is large.

The lidar return for space measurements is typically very small due to the long distance and the inversion proportional to the squared range of the lidar return signal (Eq. (A2a)). Averaging over many laser shots and/or range bins is necessary to improve the signal-to-noise ratio (SNR). For this reason, for the simulation results in Fig. 7, a horizontal average over 10 km (~5634

shots) is performed for the surface measurement, and an additional vertical average over 1 km is performed for the atmospheric measurement.

The signal-induced shot noise (the first term in the denominator in Eq. (A1)) is the dominant noise source when the lidar return signal is very strong, such as the lidar return from the land surface. In this case, the contributions from other noise sources are negligible. This is evident in the case of the surface CO2 and pressure measurement using the surface return (Figs. 7c and 7d).

Consequently, the surface measurement performs similarly during the day and night, as shown in Figs. 7e and 7f, where the nighttime and daytime simulated curves overlay each other. It is noteworthy that the solar radiation is much smaller at 1.96 µm (~220 times) than in the visible regime and approximately 2.3 times smaller than that on Earth (Table 1). Therefore, the impact of solar background noise on the daytime measurement in the near IR regime is not as significant as in the visible regime.

On the other hand, the lidar return from atmospheric scattering is very weak, and the detector noise, as well as solar background noise during the daytime measurement, play an important role. For this reason, the SNR simulated for the atmospheric return is smaller during the day than at night due to the presence of solar background noise (Fig. 7b).

**Acknowledgements**

Authors are grateful for Dr Robert Hargreaves at the Center for Astrophysics, Harvard & Smithsonian for the helpful discussion

on calculating the Martian CO2 spectroscopic characteristics using the HAPI software. This study is supported by the NASA's Planetary Instrument Concepts for the Advancement of Solar System Observations (PICASSO) Program.

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

**Figure 1: Line-by-line calculated CO$_2$ absorption spectrum of the ν′(20011) vibrational band for T = 150$^\circ$C and P = 0.006 atm using the HITRAN Application Programming Interface (HAPI) software.**

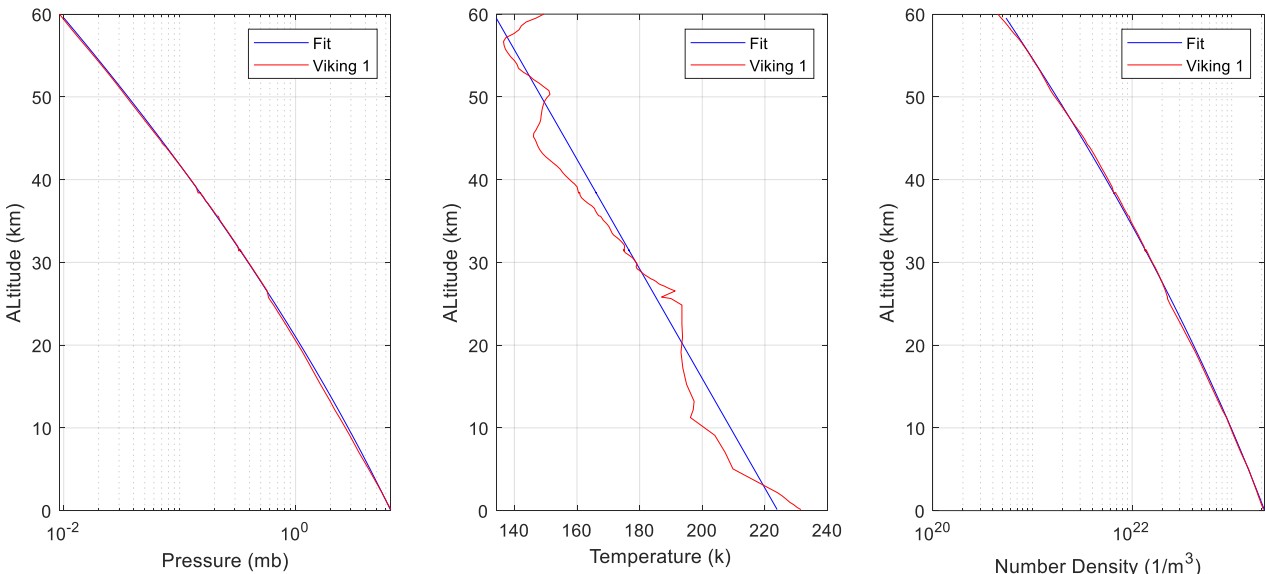

**Figure 2: Pressure (left), temperature (middle), and number density (right) profiles on Mars measured by Viking 1 (red) and fit (blue curves).**

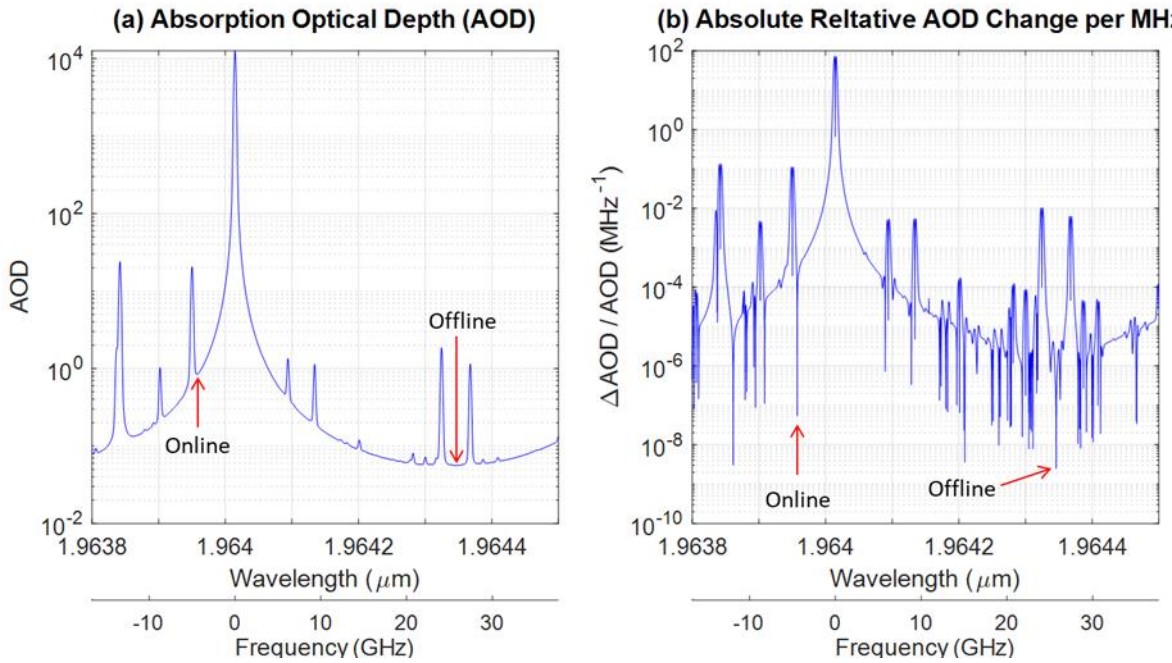

**Figure 3: (a) CO₂ absorption optical depth (AOD) using Eq. (2) from 60 km as TOA for Mars and (b) the absolute value of its change for a 1 MHz variation in laser frequency, i.e., the derivative of AOD relative to frequency. The elongated tails toward zero in (b) correspond to a trough or peak in (a). Some weak absorption lines that are not visible in (a) are enhanced in visibility in (b). The arrows indicate the selected online and offline laser wavelengths.**

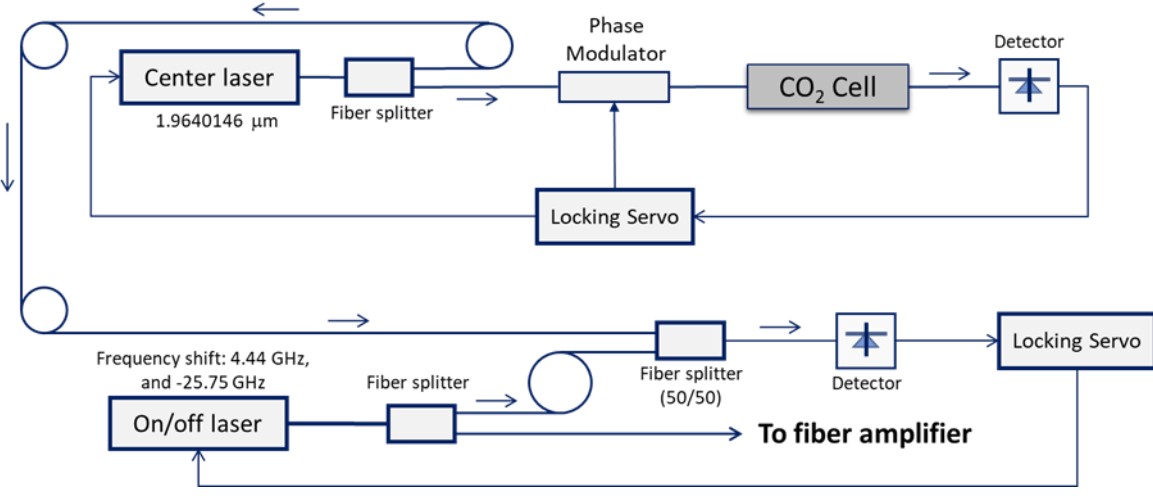

**Figure 4: Conceptual diagram for master laser wavelength locking and control.**

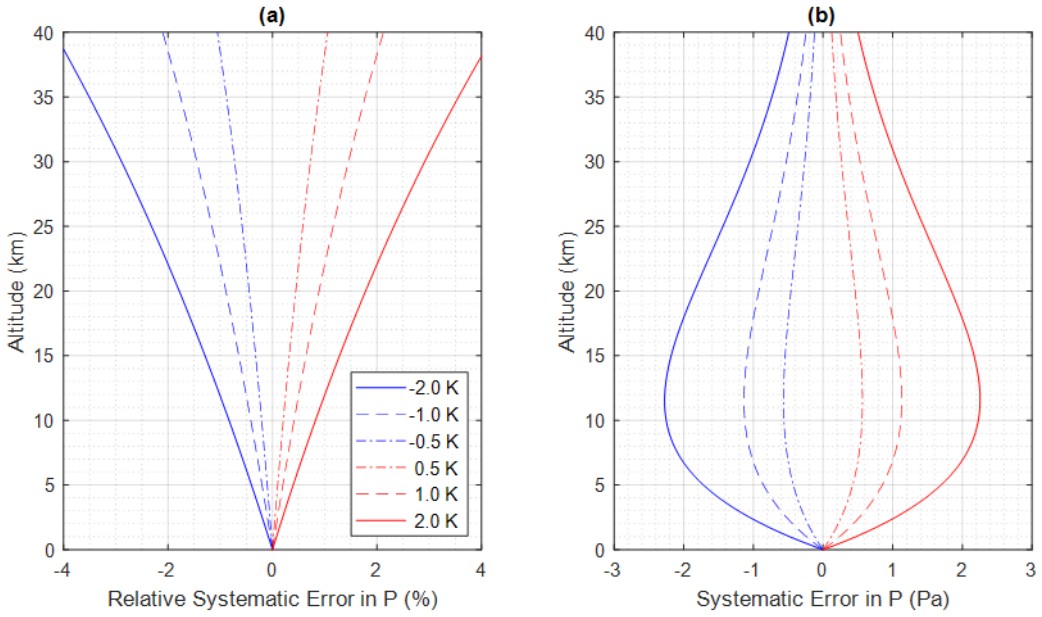

**Figure 5: (a) Relative systematic error and (b) systematic error in pressure P as a function of altitudes due to a bias of 0.5 K, 1.0 K, and 2.0 K in temperature T, calculated using Eq. (12).**

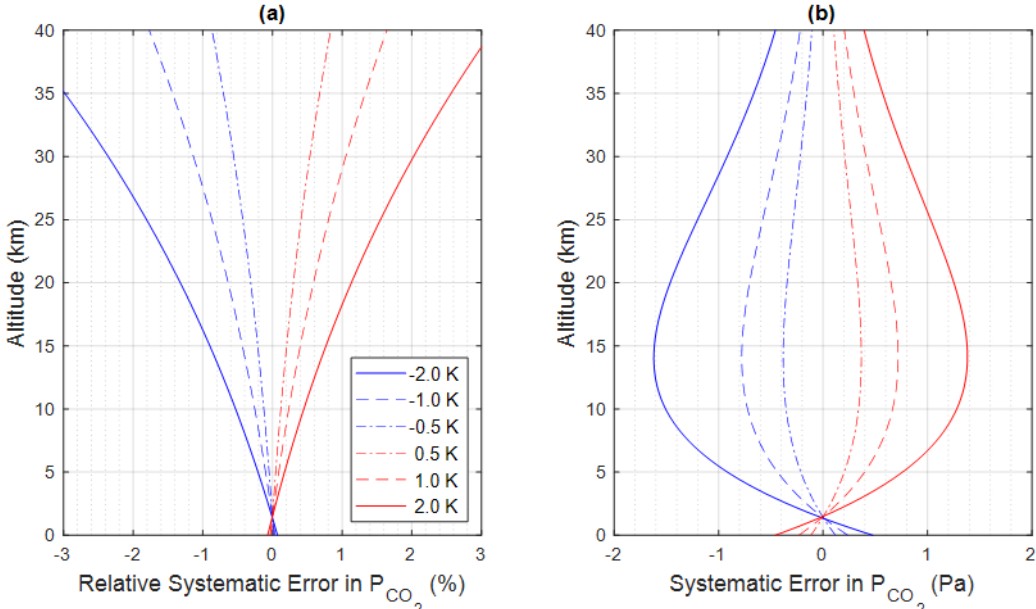

**Figure 6: (a) Relative systematic error and (b) error in $P_{CO2}$ retrieved from CO₂ DAOD due to ±0.5 K, ±1.0 K, and ±2.0 K bias in T, calculated using HAPI.**

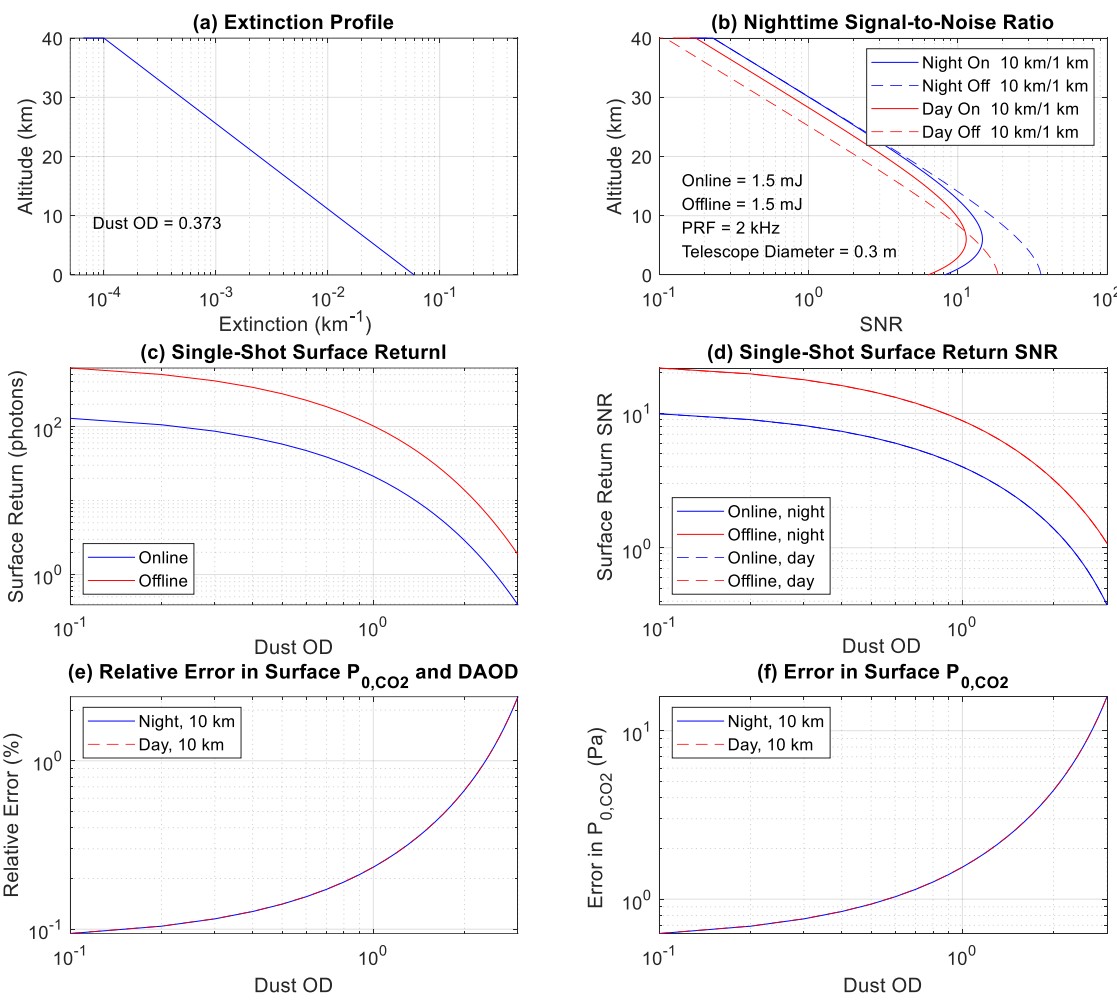

**Figure 7: (a)** Modelled dust extinction distribution, **(b)** SNRs for the online and offline atmospheric return at a horizontal resolution of 10 km and vertical resolution of 1 km during day and night, **(c)** surface return signal, and **(d)** surface return SNR during day and night, **(e)** relative errors in the $CO_2$ DAOD measurement and **(f)** errors in $P_{0,CO2}$ due to random noise. Day and night curves overlay in (d) – (f).

**Table 1 Comparisons of space lidar measurements on Mars and Earth**

|  |  | Mars 1.964 μm | CALIPSO/Earth 0.532 μm | Figure of Merit | Remark |
|---|---|---|---|---|---|
| Satellite | height (km) | 250 | 710 | 8.05 | signal ~ 1/height² |
|  | on ground speed (km/s) | 3.45 | 7.5 | 2.17 | signal ~ 1/speed |
| Photon number per mJ |  | 10.320e+27 | 2.6782e+27 | 3.85 | signal ~ $N_{photon}$ |
|  | constant (kW/m²) | 0.59 | 1.361 | 2.3 |  |

| Solar radiation | visible to IR irradiance ratio | | | ~ 220 | background noise ~ solar radiation |
|---|---|---|---|---|---|
| Atmospheric backscatter | | Dust from surface up to ~50 km | Aerosol in the low atmosphere | | signal ~ backscatter |

**Table 2 Lidar system parameters used in OSSEs**

| | | |
|---|---|---|
| Laser | pulse energy, online / offline (mJ) | 1.5 / 1.5 |
| | pulse repetition frequency (Hz) of each wavelength | 2000 |
| | pulse width (ns) | 200 |
| | beam expander throughput | 0.883 |
| | Wavelengths: online, and offline ($\mu$m) | 1.9639572, 1.9643460 |
| Telescope | diameter (m) | 0.3 |
| | clear area ratio | 0.882 |
| Detector (DRS APD) | quantum efficiency | 0.9 |
| | fill factor (that quantifies the proportion of the received return laser signal effectively incident on an APD element), | 0.75 |
| | Unity-gain dark current (A) with read-out integrated circuit | 3.5e-13 |
| | gain | 900 |
| | Excess noise factor | 1.3 |
| Lidar receiver FOV (mrad) | | 0.13 |
| Solar blocking filter bandwidth (nm) | | 0.8 |
| System optical throughput | | 0.545 |
| Sun zenith angle at equator (for daytime simulation) | | 30° |
| Surface reflectivity (sr$^{-1}$) | | 0.161 |
| Satellite altitude (km) | | 240 |