# Peer review of "Martian column CO2 and pressure measurement with spaceborne differential absorption lidar at 1.96 µm"

_Atmospheric Measurement Techniques, 2023_

## Referee Comment (RC2)

The manuscript, titled "Martian column $CO_2$ and pressure measurement with differential absorption lidar at 1.96 μm", presents a lidar instrument for measuring atmospheric carbon dioxide ($CO_2$) on Mars for surface pressure determination. The manuscript reports a similar lidar instrument, as previously presented by some of the authors referred to in (Lin and Liu, 2021), but with different transmitter using 1.9 μm laser instead of 2.05 μm. The advantages of changing the lidar operating wavelength are not clearly justified. In addition, the manuscript suffers some oversights and lacks detailed discussions in some parts, which may lead to incorrect conclusions and ambiguity to the reader. The authors are encouraged to reevaluate the work analysis and presentation for future resubmission, while addressing the following issues.

1.  The presented active lidar instrument cannot fully perform the surface pressure measurement by itself. It requires additional supporting instruments, such as infrared temperature sounder (section 2.3) for temperature profile measurement and involves iterative analysis with predetermined initial measurements. Even molecular atmospheric pressure, other that $CO_2$, requires more dedicated measurements or modeling (lines 106-108). For example, the multispectral sounder passive instrument, cited by (Natraj et al., 2022) can perform temperature, $CO_2$, and other molecules profiling, which can be used to derive the pressure on Mars, without the risk of using active element. Therefore, a strong justification for using a high-risk lidar instrument for Mars is required.
2.  The presented analysis focused on measurement errors without addressing the measurement itself. The analysis should address assessments for the $CO_2$ differential optical depth and surface pressure measurement using the technology proposed in Table 2, including surface return signals. For example, the table lists several parameters, such as telescope and detector specifications, which were not included in modeling. Other specifications, such as detector noise, digitizer specifications and signal averaging were not addressed. The error analysis is limited and does not include other significant factors, such as molecular interference, electronic noise, laser jitter, ranging uncertainty, etc. For example, see (Refaat et al., 2015).
3.  The spectral modeling presented in figures 1 and 3a focuses on strong $CO_2$ absorption lines around the suggested operating online and offline wavelengths. Nevertheless, weaker lines, within the same spectral band, should be included in the analysis since they contribute to the cross section and optical depth profiles. For example, weak lines contribution is significant toward the offline and may require updating its position. Probably this will change the results, such as the error presented in figure 3b.
4.  The lidar operating platform is not clearly specified from the beginning. Later, toward the end of the paper, specifically in Table 2, surface reflectivity and satellite altitude are given, which implies an orbiter platform while the lidar pointing nadir toward the surface. Assuming this is true, the integration limits presented in the analysis are incorrect. For an arbiter, the column integration should start from the orbiter altitude, z, to the surface (at z = 0) not to the top of the atmosphere (TOA). As a matter of fact, even the definition of TOA on Mars is unclear. This appears in equations (2), (3) and (6), while equations (8) and (12) presents the correct integration limits but flipped (e.g., integration from z to 0 not 0 to z). It is unclear how this error would change the analytical results presented in figures 5 to 7 and

conclusions. Also, the meaning of presenting the errors with altitude z, is unclear. Does that means changing the orbiter altitude?

In addition, the authors should consider the following specific details:

**Section 1: Introduction**

Line 27: Please avoid the use of the word "air" for referring to Martian atmosphere, as it is relevant to Earth. For example, living creatures on Earth breath air, and the use of this word may imply that Mars atmosphere supports life. Same applies to the rest of the document, whenever the word "air" is used to address Martian atmosphere.

Lines 27-28: It is unclear how atmospheric dynamics are specifically associated with "dry air movements" not "total air movement" on extraterrestrial planets. This implies that water vapor, the difference between total and dry air, has no role in atmospheric dynamics. Please elaborate or change.

Lines 33-35: Please check the composition of the Martian atmosphere. Martian atmosphere does not contain hydrogen. The composition ratios listed for carbon dioxide, nitrogen and argon are inconsistent to what are listed in (Franz et al., 2017 and Williams, 2020).

Lines 35-26: Please explain the difference between "global atmospheric dynamics" and "global atmospheric thermodynamics". Please elaborate on "the carbon cycle on Mars" and include a reference. Is the sentence refereeing to a current active carbon cycle or historical carbon cycle?

Line 39: Please elaborate on atmospheric temperature measurements using infrared remote sensing techniques and include reference. For example, introduce the infrared temperature sounder, referred to later in Line 66.

Lines 39-42: This paragraph addresses pressure measurement limitations on Earth. It is unclear how that is relevant to pressure measurement on Mars. Please explain.

Line 46: Please consider rephrasing "Passive instrument" to "Passive remote sensing instruments" to distinguish from in-situ barometers presented in the previous paragraph.

Line 67-69: For complete argument, it is beneficial to address some of the risks using a DIAL system on Mars, such as cost, complexity, lifetime, power consumption, weight, and size, etc., as compared to passive remote sensing.

Line 70-76: In this paragraph, please consider introducing the operating platform and target for this proposed lidar on Mars. For example, is it for a lander mission or orbiter. The last two sentences can move to section 2.4 "Wavelength Selection".

**Section 2.1: DIAL Measurement**

Line 84: Please specify that the $CO_2$ differential absorption optical depth is for a single path, as presented in equation (1). It is more common to use double-path optical depth analysis, since the transmitted radiation must travel forth and back, from and to, the lidar instrument.

Line 86: In equation (1), the measured differential optical depth, represented by the right term, includes the $CO_2$ differential optical depth and other differential optical depths from interfering molecules, as specified in (Lin and Liu, 2021 equation 5). Please specify and comment on why it is not included in this analysis.

Line 87: The sentence "$\Delta\tau_{CO2}$ represents the $CO_2$ DAOD at the online and offline wavelengths" is redundant.

Line 88-90: It is unclear why the transmitted laser energy, or power, was replaced by the calibration coefficients, and if these coefficients are assumed constants or variables in equation (1). An equation for the calibration coefficients, including units, would be helpful to show how they can be obtained through the zero-range measurement. For example, lidar equation is not defined at zero range due to the backscattered signal dependance on the reciprocal of the range squared (i.e., Lidar Signal $\rightarrow \infty$ at Range = 0).

Line 90-91: The cited references (Lin et al., 2015; Dobler et al., 2013; Campbell et al., 2020) are irrelevant since they present an intensity-modulated continuous-wave lidar systems, whereas the described system is pulsed (as pointed out in Line 71). Please check and update.

Line 92: Equation (2) is unclear since the lidar operating platform is not specified. Please see comment #4 and define the symbol z'.

Line 96: Equation (3) is redundant to equation (2) just by arranging terms. Probably it is better to solve (2) and (3), and present

$$N_{CO2}(z) = \int_{z}^{TOA} n_{CO2}(z')dz'$$

which is referred to as "the column $CO_2$ molecular number integrated from z to the top of atmosphere (TOA)". If we assume z' is altitude in meter and $n_{CO2}$ is the number density in $1/m^3$, then $N_{CO2}$ must be in $1/m^2$. Therefore, the physical interpretation of $N_{CO2}$ is unclear. Please explain.

Line 98: Equation (4) is redundant to equation (2).

Line 99: By "the air pressure caused by $CO_2$" does it means "$CO_2$ partial atmospheric pressure at the surface"?

Line 100-103: Please include references for equations (5) and (6) or derivation. Why "the weighted mean Martian gravitation acceleration" is required not Martian gravitation acceleration? Please define $n_{CO2,model}$ in equation (6).

Line 102: The weighted mean Martian gravitational acceleration between z and TOA, represented by equation (6) is different than the representation in (Lin and Liu, 2021 equation 10). Why extra denominator was included?

Line 106-107: Please quantify $P_{others}$, relative to $P_{CO2}$ here (Line 209) and validate the assumption of stability. What other "dedicated measurements" are required to measure $P_{other}$? If the plan to

send an additional instrument to measure $P_{other}$, can it measure the total pressure as well? Then what is the benefit to send a lidar? Please compare lidar to (Natraj et al., 2022) for justification.

**Section 2.2: Surface Column $CO_2$ and Pressure Measurement**

Line 109: For the title of this section, is it meant to be "Column $CO_2$ and Surface Pressure Measurement"?

Section 2.2 is too short compared to other sections. Consider combining with previous section.

**Section 2.3: Atmospheric Pressure Measurement with IR Sounder Temperature Measurement**

Line 126-128: Please include a reference for equation (8) and the [average] molar Mass of Martian atmosphere. Define zero altitude on Mars used for the integration limits. Can the barometric formula be applied on Mars?

Line 131-132: This is confusing. Please state the difference between equations (8) and (5). Is one equation for surface pressure and the other for pressure profile? This indicates that an initial pressure profile is required to measure the pressure profile.

Line 139-140: Generally, iterative processes may converge or diverge. It is not clear if surface pressure determination through iterative process would converge. Please comment.

**Section 2.4: Wavelength Selection**

Line 158: Is it the Absorption Optical Depth (AOD) required to be 1.1 or the Differential Absorption Optical Depth (DAOD), as claimed in Line 145?

Line 164-165: How many HITRAN lines were used for AOD calculations and the criteria for selecting these lines? The results of Figure 3a indicates that weaker lines were neglected, which significantly contribute to the spectral profile. Please investigate since this may change your conclusions, such as the required online and offline positions and laser line stability.

Line 172-173: Please mark P(12) line on figure 1. What about other lines presented in figure, how do they compare to the selected P(10) line? Otherwise, limit the figure to the discussed lines.

**Section 2.5: Laser and Wavelength Locking**

Line 183-184: How the DAOD error, of less than $10^{-4}$, was evaluated?

**Section 3.1: Error Analysis**

Line 193-198: Please define all symbols and discuss these equations.

Line 199: Which results are referred to? Do you mean "analysis"?

Line 206-207: Please include the error due to the initial pressure estimate used for the iterative process.

Line 209: Please include reference for <5% other pressure.

**Section 3.2: Simulation Results**

Line 238: Please define the figure of merit (FOM).

Line 240-241: Please state how the photon number per pulse was evaluated? Why was the photon number addressed here not signal as equation (1)?

Line 245: Typically, signal-induced shot noise is the dominant noise source for lidar systems, whereas daytime background can limit the dynamic range. Moreover, background blocking filters can resolve this issue as pointed out in (Line 294). Please investigate.

Line 277: Please elaborate on solar background noise calculation and why it wasn't included in the error analysis presented in equations (9) to (12).

Line 289-291: Need a figure for simulating the lidar return signals and $CO_2$ DAOD and surface pressure retrievals to support these claims.

**Figures and Tables**

Line 400: Figure 1: Please indicate the spectral resolution of the calculated cross section. It is unclear why wide spectral range is shown, rather than focusing on 1.9640146 μm line. This calculation focuses on the dominant spectral lines while ignoring weaker lines. Please include weaker line and plot in log scale to be comparable to the optical depth calculations presented in Figure 3.

Line 406: Figure 2: In addition, please include typical Martial vertical $CO_2$ profile as applied to equation (6).

Line 408: Figure 3: How the absorption optical depth, presented in figure 3a, was calculated? Is it based on a model similar to equation (2)? Please state the altitude limits.

Line 416: Figure 6: Please replace "system" with systematic". Check if these profiles calculated using HAPI or equations (11).

Line 425: Table 2: Some of the parameters listed in this table were not discussed within the manuscript. Please include a discussion for how these parameters are relevant to the measurements. For example, the beam expander throughput, telescope diameter and clear area ratio, detector quantum efficiency and dark current. What is the meaning of the "fill factor"? is the DRS APD a single detector or array? What is the detector noise-equivalent-power and how it influences the errors? Please define abbreviations.

---

## Author Comment (AC1)

This study introduced a design of lidar to retrieve Martian column $CO_2$ and pressure profiles. Compared with previous study, the main innovation is compact lidar configuration with small telescope dimensions and low laser pulse energies using the 1.96 μm $CO_2$ absorption band. Online and offline bands around 1.96 μm are selected to formulate a differential absorption optical depth algorithm. Finally, the authors conducted an OSSE experiment to estimate retrieval uncertainties. When the design is implemented, retrieved data should be useful for Martian research. The manuscript is well written.

We appreciate the time and effort you have dedicated to reviewing our manuscript. Your insights, along with those of the other reviewer, have been valuable. We have thoroughly revised the manuscript, and our responses to your comments are provided below in blue.

Specific comments:

In Fig. 2, temperature data is not well fitted. Is the fitted temperature curve used for calculation in Fig.3? If so, how it affects $CO_2$ absorption optical depth in Fig. 3?

There are few percent errors in the fitted temperature curve. These errors exhibit some random nature, and the overall error for the entire profile is not significant. They do not substantially impact the $CO_2$ number density, as demonstrated in the newly added Fig. 3c, which directly influences the $CO_2$ absorption optical depth (AOD). Therefore, the impact of the temperature fitting error on $CO_2$ AOD is not significant. We note that the curve is used for simulation purposes only, and both pressure and temperature data should be updated in the future in the data processing.

Technical correction:

Lines 305 and 321: 2 should be subscript in CO2.

Corrected.

---

## Author Comment (AC2)

The manuscript, titled 'Martian column $CO_2$ and pressure measurement with differential absorption lidar at 1.96 μm', presents a lidar instrument for measuring atmospheric carbon dioxide ($CO_2$) on Mars for surface pressure determination. The manuscript reports a similar lidar instrument, as previously presented by some of the authors referred to in (Lin and Liu, 2021), but with different transmitter using 1.9 μm laser instead of 2.05 μm. The advantages of changing the lidar operating wavelength are not clearly justified. In addition, the manuscript suffers some oversights and lacks detailed discussions in some parts, which may lead to incorrect conclusions and ambiguity to the reader. The authors are encouraged to reevaluate the work analysis and presentation for future resubmission, while addressing the following issues.

We are grateful for the thorough review and insightful comments provided by the reviewer. We sincerely appreciate the time and effort dedicated to evaluating our work. While the feedback has been constructive and has contributed significantly to the improvement of the manuscript, we acknowledge that certain major points raised by the reviewer may have resulted from potential misunderstanding or speculation about the content. This could be attributed to the broad technical areas covered in the manuscript, spanning science, observational environment, instrumentation, and spectroscope, as well as the exclusion of some common details initially deemed unnecessary or beyond the paper's scope. Unfortunately, these misunderstandings and speculations seem to have been treated as factual bases in the review, leading to suggestions for a reevaluation of the work.

In response, we have made improvements to the manuscript by incorporating a more comprehensive description, aiming to better cater to a broader audience. Additionally, we have added an appendix detailing the Signal-to-Noise Ratio (SNR) calculation. Detailed responses to each comment are appended in blue color. We are committed to enhancing the clarity and accuracy of our work based on the valuable insights provided by the reviewer.

1. The presented active lidar instrument cannot fully perform the surface pressure measurement by itself. It requires additional supporting instruments, such as infrared temperature sounder (section 2.3) for temperature profile measurement and involves iterative analysis with predetermined initial measurements. Even molecular atmospheric pressure, other that $CO_2$, requires more dedicated measurements or modeling (lines 106-108). For example, the multispectral sounder passive instrument, cited by (Natraj et al., 2022) can perform temperature, $CO_2$, and other molecules profiling, which can be used to derive the pressure on Mars, without the risk of using active element. Therefore, a strong justification for using a high-risk lidar instrument for Mars is required.

The issue raised in this comment is a typical example illustrating that the original manuscript may lack details about DIAL surface pressure measurements, or the reviewer may not be familiar with certain aspects of lidar remote sensing. In the context of Martian surface pressure measurements, it's essential to clarify that the $CO_2$ DIAL system measurement does not require additional instruments to sound temperature profiles. Through the climatological weighting function and total $CO_2$ amount measurements from DIAL, Martian surface pressure can be effectively retrieved. The temperature sounding primarily aids in atmospheric pressure profiling based on the barometric formula. An additional benefit of the temperature sounder is its potential to improve retrieval accuracy.

It's important to note that passive remote sensing also employs a similar concept to measure Martian surface pressure through $CO_2$ observations, as indicated in the references provided in the manuscript.

Other trace gases contribute only a small fraction to surface pressure, and their variations result in negligible changes due to their high-order small numbers. Similar to passive remote sensing, it is not necessary to simultaneously measure the partial pressure due to other gases on the same platform.

The $CO_2$ DIAL is introduced in Lin and Liu (2021) and this paper to provide supplementary measurements where passive remote sensors may be limited. Its significance is particularly notable for understanding pressure and dynamics in the nighttime side and polar regions of Mars. The observations in polar $CO_2$ can provide insights into atmospheric $CO_2$ variations, dry ice formation, and their connections with polar ice changes. While spaceborne passive sensors are valuable for surface pressure observations, their limitation lies in obtaining pressures only during the daytime using solar infrared spectra (such as the 2-µm $CO_2$ absorption band), which restricts their capabilities, especially in polar regions.

In the revised manuscript we added 'However, it is important to note that a simultaneous temperature profile measurement is not mandatory in DIAL surface pressure measurements, although it can enhance measurement accuracy.' (Lines 127-128)

Climatological temperature and pressure data are commonly utilized in computing the weighting function A for Earth DIAL $CO_2$ measurements, achieving precision in the range of a few parts per million (please refer to the referenced papers for an overview). Simultaneous measurements from a thermal infrared sounder can enhance temperature profiles and contribute to improved $CO_2$ and pressure retrieval. While there is anticipation that a cost-effective thermal infrared sounder could be feasibly deployed on an orbital platform around Mars, it is important to note that such integration is not deemed necessary.

2. The presented analysis focused on measurement errors without addressing the measurement itself. The analysis should address assessments for the CO2 differential optical depth and surface pressure measurement using the technology proposed in Table 2, including surface return signals. For example, the table lists several parameters, such as telescope and detector specifications, which were not included in modeling. Other specifications, such as detector noise, digitizer specifications and signal averaging were not addressed. The error analysis is limited and does not include other significant factors, such as molecular interference, electronic noise, laser jitter, ranging uncertainty, etc. For example, see (Refaat et al., 2015).

In response to the reviewer's comment regarding the analysis focusing on measurement errors without addressing the measurement itself, we would like to clarify that a comprehensive description of $CO_2$ and pressure measurement on Mars is provided in Section 2.2, including its theoretical basis and equations.

Upon a more detailed analysis of the reviewer's other comments, it appears that the reviewer's concern about "without addressing the measurement itself" stems from the absence of an explicit equation for calculating the signal-to-noise ratio (SNR). It is crucial to emphasize that all parameters in Table 2 impact the SNR in Eq. 9, subsequently influencing measurement precision, as described in Section 2.3 and simulated in Section 3, although the SNR equation is not presented in the original manuscript.

The decision to exclude the explicit equation for calculating SNR for lidar return signals was based on its status as a well-established formula by the lidar community, commonly employed in lidar simulations and performance evaluation (e.g., Liu et al., 2000; Powell et al., 2006; Lin et al., 2013). This decision aligns with the primary objective of this paper, which is to assess the feasibility of DIAL $CO_2$ and pressure

measurement on Mars using the parameters in Table 2. However, the reviewer's concern highlights the need for the inclusion of the SNR equation for non-lidar readers. Therefore, we have introduced Appendix A in the revised manuscript. This appendix provides a detailed description of the SNR calculation equation for the benefit of general readers.

It is important to highlight that all parameters listed in Table 2, encompassing telescope and detector specifications, including detector noises, were utilized in the SNR calculation. This utilization was explicitly mentioned in the original manuscript (line 249), which the reviewer may have overlooked.

Regarding the comment on the limited error analysis and exclusion of significant factors such as molecular interference, electronic noise, laser jitter, and ranging uncertainty, it is essential to note that laser frequency stabilization, which is key for high precision observation, is addressed in Section 2.5 of the original manuscript and Section 2.4 in the revised manuscript. Furthermore, laser power jitter will be readily monitored and normalized following standard DIAL measurement practices (similar to those used in other backscatter lidars such as the CALIPSO lidar launched in 2006), as outlined in our previous publication (Lin and Liu, 2021).

In response to the term 'molecular interference' from the reviewer's comments, we interpret it as potential absorption from trace gases. It is crucial to emphasize that the $CO_2$ line was deliberately chosen to minimize absorption from other gases, ensuring their impact is negligible. To clarify this, we have added the criterion 'and (4) there is no absorption from other gases' to the wavelength selection criteria (line 169).

Furthermore, it needs to point out that a DIAL system utilizes single-frequency lasers, and the laser frequency can be stabilized to sub-MHz levels, allowing it to be set precisely at the desired location on one side of an absorption line. The authors discussed this in the original manuscript.

Z. Liu, P. Voelger, N. Sugimoto, 'Simulations of the observation of clouds and aerosols with the Experimental Lidar in Space Equipment system' Appl. Opt., Vol. 39, No. 18, 2000.

K. A. Powell, Z. Liu**,** and W. H. Hunt, 'Simulation of Random Electron Multiplication in CALIPSO Lidar Photomultipliers', 23rd International Laser Radar Conference (ILRC), Nara, Japan, 2006.

B. Lin, S. Ismail, F. W. Harrison, et al. 'Modeling of intensity-modulated continuous-wave laser absorption spectrometer systems for atmospheric $CO_2$ column measurements', Applied Optics, 52, 7062–7077, 2013.

3. The spectral modeling presented in figures 1 and 3a focuses on strong CO2 absorption lines around the suggested operating online and offline wavelengths. Nevertheless, weaker lines, within the same spectral band, should be included in the analysis since they contribute to the cross section and optical depth profiles. For example, weak lines contribution is significant toward the offline and may require updating its position. Probably this will change the results, such as the error presented in figure 3b.

We conducted a calculation line by line using HAPI, as explicitly indicated in line 163 and Caption 1 of the original manuscript. In our calculations, we considered all $CO_2$ absorption lines, encompassing both 'weaker' and 'stronger' lines of all $CO_2$ isotopes, within a large wavenumber range of 5063 - 5128$cm^{-1}$. The calculation utilized a wavenumber resolution of $5x10^{-5}$ $cm^{-1}$ (~$2x10^{-5}$ nm). The inclusion of weaker lines is evident, as shown in Fig. 3. The reviewer's assertion that 'The spectral modeling presented in

figures 1 and 3a focuses on strong $CO_2$ absorption lines around the suggested operating online and offline wavelengths' is clearly speculative and likely based on a misunderstanding of Figs. 1 and 3 (please refer to the responses to the specific comments for more details). Surprisingly, this speculation seems to have been taken as an important factual basis by the reviewer to draw conclusions in the review without attempting to verify with the authors.

4. The lidar operating platform is not clearly specified from the beginning. Later, toward the end of the paper, specifically in Table 2, surface reflectivity and satellite altitude are given, which implies an orbiter platform while the lidar pointing nadir toward the surface. Assuming this is true, the integration limits presented in the analysis are incorrect. For an arbiter, the column integration should start from the orbiter altitude, z, to the surface (at z = 0) not to the top of the atmosphere (TOA). As a matter of fact, even the definition of TOA on Mars is unclear. This appears in equations (2), (3) and (6), while equations (8) and (12) presents the correct integration limits but flipped (e.g., integration from z to 0 not 0 to z). It is unclear how this error would change the analytical results presented in figures 5 to 7 and conclusions. Also, the meaning of presenting the errors with altitude z, is unclear. Does that means changing the orbiter altitude?

Indeed, the platform is not explicitly specified in the paper; we appreciate the reviewer's observation. To address this, we added the phrase 'spaceborne' to the title, the first sentence in the abstract and introduced the clarification 'For the spaceborne DIAL measurement,' at the beginning of the sentence before Eq. (2).

We acknowledge the reviewer's observation regarding the interpretation of the integration in equations (2), (3), and (6) as extending from the lidar/platform down to the top of the atmosphere (TOA), although the observation appears to be a misunderstanding of these equations. It is crucial to clarify that in our manuscript, the variable z represents altitude, as indicated in line 93 of the original manuscript. This indicates that the integration should extend from the surface (z = 0) or the atmosphere at altitude z up to the TOA. Integrating from the platform to TOA, as questioned by the reviewer, would not yield any meaningful results.

To provide a clearer explanation of the top of atmosphere (TOA), we have made an addition to the revised manuscript: 'the top of atmosphere (TOA) which refers to the outer boundary of Mars's atmosphere. In the simulation for this study, the upper limit of the integral is set at 60 km, where the pressure is approximately 0.14% of that at the surface.' (Lines 97-99)

In addition, the authors should consider the following specific details:

**Section 1: Introduction**

Line 27: Please avoid the use of the word 'air' for referring to Martian atmosphere, as it is relevant to Earth. For example, living creatures on Earth breath air, and the use of this word may imply that Mars atmosphere supports life. Same applies to the rest of the document, whenever the word 'air' is used to address Martian atmosphere.

Thank you!

We replaced 'air' with 'atmosphere', 'atmospheric' or 'Martian atmosphere' throughout the paper.

Lines 27-28: It is unclear how atmospheric dynamics are specifically associated with 'dry air movements' not 'total air movement' on extraterrestrial planets. This implies that water vapor, the difference between total and dry air, has no role in atmospheric dynamics. Please elaborate or change.

For the Martian atmosphere, especially in the lower part, dry air movement refers to the total air movement since there is almost no water vapor. This 'dry air movement' is contrasted with 'wet air movements' which would involve latent heat release and produce moist convection, similar to what occurs on Earth. In response to the feedback, we replaced 'dry air' with 'air' to avoid unnecessary confusion for general readers.

Lines 33-35: Please check the composition of the Martian atmosphere. Martian atmosphere does not contain hydrogen. The composition ratios listed for carbon dioxide, nitrogen and argon are inconsistent to what are listed in (Franz et al., 2017 and Williams, 2020). Lines 35-26: Please explain the difference between 'global atmospheric dynamics' and 'global atmospheric thermodynamics'. Please elaborate on 'the carbon cycle on Mars' and include a reference. Is the sentence refereeing to a current active carbon cycle or historical carbon cycle?

Deleted 'hydrogen'. Deleted Franz et al and replaced numbers with the ones in Williams, 2020.

Dynamics refers to mechanically forced processes, whereas thermodynamics encompasses processes induced by thermal forcings, including radiative heating. A broader interpretation of dynamics inherently includes thermodynamics. For the sake of simplicity, the term 'thermodynamics' has been omitted in this version. The carbon cycle encompasses diverse components, encompassing fluctuations in atmospheric $CO_2$ levels over annual and longer durations, attributed to cold region dry ice deposition, sublimation, and other related processes. Seasonal variations in surface pressure are intricately linked to the carbon cycle. Additionally, a reference from Spiga et al. (2018) has been added (line 36).

Spiga, A., D. Banfield, N. Teanby, and coauthors: Atmospheric Science with InSight, Space Sci. Rev., 214:109, https://doi.org/10.1007/s11214-018-0543-0, 2018.

Line 39: Please elaborate on atmospheric temperature measurements using infrared remote sensing techniques and include reference. For example, introduce the infrared temperature sounder, referred to later in Line 66.

A historical record exists of Martian atmospheric temperature soundings. Notably, the TES (Thermal Emission Spectrometer) instrument effectively captured Mars's infrared emissions and was well-suited for retrieving the atmospheric thermal structure (Smith et al., 2001). We delve into this topic further in the current revision. 'The global measurement of planetary atmospheric temperature can be achieved through infrared (IR) remote sensing techniques. Martian atmospheric temperature soundings have a historical precedence, exemplified by instruments such as the spaceborne TES (Thermal Emission Spectrometer). This instrument effectively captured infrared emissions from Mars, proving well-suited for retrieving the atmospheric thermal structure (Smith et al., 2001).' (Lines 39-43)

Smith, M. D., Pearl, J. C., Conrath, B. J., and Christensen, P. R. Thermal emission spectrometer results: Mars atmospheric thermal structure and aerosol distribution. Journal of Geophysical Research: Planets, 106, 23929, 2001.

Lines 39-42: This paragraph addresses pressure measurement limitations on Earth. It is unclear how that is relevant to pressure measurement on Mars. Please explain.

Deleted.

Line 46: Please consider rephrasing 'Passive instrument' to 'Passive remote sensing instruments' to distinguish from in-situ barometers presented in the previous paragraph.

Done.

Line 67-69: For complete argument, it is beneficial to address some of the risks using a DIAL system on Mars, such as cost, complexity, lifetime, power consumption, weight, and size, etc., as compared to passive remote sensing.

Thank you! It is a valid and fair point to consider addressing potential issues associated with the use of the DIAL system on Mars. We have incorporated additional information, stating that 'It is noteworthy that deploying active optical remote sensing instruments for space measurements typically incurs higher costs compared to passive ones. Therefore, adopting a strategy that involves utilizing active remote sensing instruments with support from lightweight, compact, and low-cost passive remote sensing instruments would be advantageous. This approach is exemplified in the CALIPSO mission, where the payload includes one active backscatter lidar, infrared imaging radiometer, and a wide-field camera (Hunt et al., 2009).' (lines 68-72).'

Line 70-76: In this paragraph, please consider introducing the operating platform and target for this proposed lidar on Mars. For example, is it for a lander mission or orbiter. The last two sentences can move to section 2.4 'Wavelength Selection'.

Great suggestion! Thanks! We inserted 'using a $CO_2$ DIAL on an orbiter'. We also moved the last two sentences to section 2.4 as suggested.

**Section 2.1: DIAL Measurement**

Line 84: Please specify that the $CO_2$ differential absorption optical depth is for a single path, as presented in equation (1). It is more common to use double-path optical depth analysis, since the transmitted radiation must travel forth and back, from and to, the lidar instrument.

Thanks for the comment! However, single-path (usually called one-way) optical depth is commonly used by the lidar community. Added 'one-way' before '$CO_2$ differential absorption optical path' (line 85).

Line 86: In equation (1), the measured differential optical depth, represented by the right term, includes the $CO_2$ differential optical depth and other differential optical depths from interfering molecules, as specified in (Lin and Liu, 2021 equation 5). Please specify and comment on why it is not included in this analysis.

When online and offline are closely positioned, the other differential optical depths can be ignored, as pointed out in Lin and Liu (2021). We rewrote the sentence before Eq. (1) as 'The selected online and offline wavelengths are closely positioned, resulting in negligible differences in optical depths due to attenuations other than $CO_2$ absorption (mainly scattering optical depths) between these wavelengths. Therefore, the one-way $CO_2$ differential absorption optical depth (DAOD) can then be determined by calculating the ratio of received signals at the online and offline wavelengths (Lin and Liu, 2021):'

Line 87: The sentence '$\Delta\tau_{CO_2}$ represents the $CO_2$ DAOD at the online and offline wavelengths' is redundant.

Deleted

Line 88-90: It is unclear why the transmitted laser energy, or power, was replaced by the calibration coefficients, and if these coefficients are assumed constants or variables in equation (1).

An equation for the calibration coefficients, including units, would be helpful to show how they can be obtained through the zero-range measurement. For example, lidar equation is not defined at zero range due to the backscattered signal dependance on the reciprocal of the range squared (i.e., Lidar Signal $\rightarrow$ $\infty$ at Range = 0).

$C_0$ should be the system constant containing system parameters and other range-independent quantities which may vary over time. We replaced 'calibration coefficient' with 'system constant' (line 89). We also added Appendix A to provide more detailed equations used in the SNR calculation.

The term 'zero-range' does not necessarily mean using signals at range = 0 but rather refers to the technique that uses a range very close to the lidar, so that the $CO_2$ absorption is negligibly small. We revised the sentence concerning zero-range calibration, and the updated statement is as follows: 'This ratio can be determined by the ratio of online and offline signals backscattered from a target close to the lidar or a target where the $CO_2$ absorption is effectively zero.' (Lines 91-93)

Line 90-91: The cited references (Lin et al., 2015; Dobler et al., 2013; Campbell et al., 2020) are irrelevant since they present an intensity-modulated continuous-wave lidar systems, whereas the described system is pulsed (as pointed out in Line 71). Please check and update.

Regarding DIAL calibration, it does not matter whether it is pulsed or continuous wave system. This type of CW system uses a swept frequency technique where one uses a matched filter to transform the return signal into pulses. Therefore the process of deriving the CO2 absorption is the same.

Line 92: Equation (2) is unclear since the lidar operating platform is not specified. Please see comment #4 and define the symbol z'.

z is altitude as defined both in the original and the revised manuscript.

Line 96: Equation (3) is redundant to equation (2) just by arranging terms.

While Eq. (3) is derived from Eq. (2), it defines the weighting function A, akin to the kernel function in passive measurements. This parameter is crucial for data processing and must be clearly defined. Therefore, we have retained it in the formulation.

Probably it is better to solve (2) and (3), and present

$$N_{CO_2}(z) = \int_Z^{TOA} n_{CO_2}(z')dz'$$

which is referred to as 'the column $CO_2$ molecular number integrated from z to the top of atmosphere (TOA)'. If we assume z' is altitude in meter and $n_{CO_2}$ is the number density in $1/m^3$, then $N_{CO_2}$ must be in $1/m^2$. Therefore, the physical interpretation of $N_{CO_2}$ is unclear. Please explain.

$N_{CO2}$ is the column number density of $CO_2$, measured in units of $1/m^2$ as the reviewer derived, indicating the number of $CO_2$ molecules per unit area along a vertical column in the atmosphere. The column number density is a well-known parameter, particularly relevant in atmospheric science and remote sensing. The variables $z$ and $z'$ denote altitudes, as specified in both the original and revised manuscript.

Certainly, it would be preferable to solve Eqs. (2) and (3) if an analytical solution were available. This could be achieved by a layer-by-layer approach from TOA down to the bottom layer when Differential Optical Absorption Spectroscopy (DOAS) profile ($\Delta\tau$) is measured, as demonstrated in Lin and Liu (2021). However, this is not applicable in the current study due to the reduced laser energy and telescope size. Therefore, we could employ an iterative approach to solve Eqs. (2) - (8) due to the unavailability of lidar profile data.

Line 98: Equation (4) is redundant to equation (2).

Again, this is the equation would be used in the data processing like Eq. (3). Therefore, it is necessary.

Line 99: By 'the air pressure caused by $CO_2$' does it means '$CO_2$ partial atmospheric pressure at the surface'?

Yes, thanks, it does mean '$CO_2$ partial atmospheric pressure at z or the surface'. The change has been made.

Line 100-103: Please include references for equations (5) and (6) or derivation. Why 'the weighted mean Martian gravitation acceleration' is required not Martian gravitation acceleration? Please define $n_{CO2,model}$ in equation (6).

Equation 5 is formulated based on the definition of atmospheric pressure, representing the cumulative weight of molecules in a vertical column per unit area on the surface. It does not necessitate a reference point. Equation 6 elaborates on the weighted mean gravitational acceleration introduced in Eq. 5. For the sake of simplicity, we incorporate the weighted mean gravitational acceleration into the equation, acknowledging its variability based on latitude and altitude.

$n_{CO2}$ is defined in the revised manuscript (line 95).

Line 102: The weighted mean Martian gravitational acceleration between z and TOA, represented by equation (6) is different than the representation in (Lin and Liu, 2021 equation 10). Why extra denominator was included?

In statistics, Eq. (6) is the general form used to calculate the weighted mean for a quantity (here is g) with a distribution like $n_{CO2}$. The denominator was omitted in Lin and Liu (2021) because the normalized number density was used. We have addressed this omission in the revised manuscript.

Line 106-107: Please quantify $P_{others}$, relative to $P_{CO2}$ here (Line 209) and validate the assumption of stability. What other 'dedicated measurements' are required to measure $P_{other}$? If the plan to send an additional instrument to measure $P_{other}$, can it measure the total pressure as well? Then what is the benefit to send a lidar? Please compare lidar to (Natraj et al., 2022) for justification.

$P_{CO2}$ on Mars exhibits variability across various spatiotemporal scales influenced by atmospheric dynamics and the dynamic deposition and sublimation of $CO_2$, particularly at the poles. In contrast, Pother remains relatively stable. According to observations by Trainer et al. (2019), the variations in

trace gases on Mars are notably small. The annual mean volume mixing ratios for the atmosphere at their observational site are $N_2$ = 0.0259 ($\pm$0.0006), Ar = 0.0194 ($\pm$0.0004), $O_2$ = 1.61 ($\pm$0.09) x $10^{-3}$, and CO = 5.8 ($\pm$0.8) x $10^{-4}$. Consequently, the contribution of uncertainty in Pother to $P_{total}$ is minimal due to its small fraction. For instance, a 3% uncertainty in $N_2$ would contribute less than 0.15% uncertainty to $P_{total}$. $P_{other}$ can be determined through climatology or model simulations, and simultaneous measurements are not considered essential. This perspective is reinforced by referencing the study of Trainer et al. (Line 113)

Trainer, M. G., Wong, M. H., McConnochie, T. H., Franz, H. B., Atreya, S. K., Conrad, P. G., et al. (2019): Seasonal Variations in Atmospheric Composition as Measured in Gale Crater, Mars. *Journal of Geophysical Research: Planets, 124*, 3000–3024. https://doi.org/10.1029/2019JE006175.

We found the wording 'dedicated measurements' to be misleading. We changed 'other dedicated measurements' to 'other available measurements.' (Line 114)

**Section 2.2: Surface Column $CO_2$ and Pressure Measurement**

Line 109: For the title of this section, is it meant to be 'Column $CO_2$ and Surface Pressure Measurement'?

Thanks! Should be 'Column $CO_2$ and Surface Pressure Measurement', but the section is merged to 2.1 in the revised manuscript.

Section 2.2 is too short compared to other sections. Consider combining with previous section.

Thanks! Merged it to 2.1, as suggested.

**Section 2.3: Atmospheric Pressure Measurement with IR Sounder Temperature Measurement**

Line 126-128: Please include a reference for equation (8) and the [average] molar Mass of Martian atmosphere. Define zero altitude on Mars used for the integration limits. Can the barometric formula be applied on Mars?

This equation was formulated based on the hydrostatic condition and is widely applicable to planetary atmospheres. The authors do not believe there is a necessity for a reference. However, if one is desired, the textbook by Wallace and Hobbs could serve as a suitable option. Apart from the parameter adjustments in the equation, a key distinction between Mars and Earth lies in the treatment of temperature. For Earth, the temperature should be considered as virtual temperature, accounting for water vapor and potential latent heat release. In contrast, the dry atmosphere of Mars, particularly at the altitudes relevant to this study, simplifies this aspect, requiring consideration only of temperature without factoring in water vapor.

Line 131-132: This is confusing. Please state the difference between equations (8) and (5). Is one equation for surface pressure and the other for pressure profile? This indicates that an initial pressure profile is required to measure the pressure profile.

Equation (5) defines the relationship between number density and pressure, while Equation (8) represents the barometric formula. If the $CO_2$ number density is measured for each layer by DIAL, as detailed in Lin and Liu (2021), the $CO_2$ partial pressure profile can be directly obtained from Eq. (5). However, for the present study, due to the noisier atmospheric measurements resulting from a reduced telescope size and laser energy, only the column $CO_2$ can be measured using the surface return signal for

moderate dust loading. In this scenario, only the surface pressure can be directly measured using DIAL, but the pressure profile can still be retrieved using the barometric formula if the temperature profile is known, modeled, or measured.

Line 139-140: Generally, iterative processes may converge or diverge. It is not clear if surface pressure determination through iterative process would converge. Please comment.

The equations have a unique solution, and an iterative procedure is employed to derive that solution. The iterative process converges in this context.

**Section 2.4: Wavelength Selection**

Line 158: Is it the Absorption Optical Depth (AOD) required to be 1.1 or the Differential Absorption Optical Depth (DAOD), as claimed in Line 145?

DAOD is required to be 1.1.

Line 164-165: How many HITRAN lines were used for AOD calculations and the criteria for selecting these lines? The results of Figure 3a indicates that weaker lines were neglected, which significantly contribute to the spectral profile. Please investigate since this may change your conclusions, such as the required online and offline positions and laser line stability.

Please also refer to the response to the general comment above. Essentially all absorption lines (6641 lines in total) from all $CO_2$ isotopes are considered. We added the following sentences to provide more details to describe the details of our HAPI calculation. 'All absorption lines of all $CO_2$ isotopes within the wavenumber range of 5063 - 5128 $cm^{-1}$, covering the entire absorption band, are considered in the HAPI line-by-line calculation. The calculation is performed with a wavelength resolution of $5x10^{-5}$ $cm^{-1}$ (equivalent to ~$2x10^{-5}$ nm), utilizing the Voigt distribution with a wing length of 10 $cm^{-1}$.' (Lines 176-179)

Addressing the reviewer's statement about 'neglected weaker lines,' we would like to clarify that, while we are uncertain about the specific weaker lines referred to by the reviewer, it is important to note that all $CO_2$ lines have been incorporated in both Figs. 3a and 3b. There is no justified rationale for excluding any of these lines from our analysis. It is puzzling how the reviewer drew this assertion from Fig. 3a, but we hypothesize that this inference may have arisen from a comparison between Fig. 3a and 3b due to the different visibility of weak and strong lines in these figures. It is essential to clarify that Fig. 3b is essentially a derivative of Fig. 3a. The enhanced visibility of weaker lines in Fig. 3b can be attributed to the fact that the derivative of the troughs or peaks approaches zero. When plotted on a logarithmic scale, this results in an elongated tail toward zero in Fig. 3b. The reviewer's misinterpretation of Fig. 3, coupled with other speculations mentioned earlier, appears to have influenced on the conclusion of the reevaluation of the analysis.

To improve the understanding of Fig. 3 for general readers, we have modified the sentence to convey, 'Figure 3b, essentially a derivative of Fig. 3a, illustrates that the sensitivity of AOD to laser frequency variability is significantly smaller in the trough regions compared to the surrounding areas. This is because the derivative or slope at the trough region is close to zero.' (Lines 179-181) Additionally, we have expanded the caption for Fig. 3 to provide more comprehensive information. The revised caption now reads as follows: '(a) $CO_2$ absorption optical depth (AOD) for Mars, and (b) the absolute value of its change for 1 MHz variation in laser frequency, i.e., the derivative of AOD relative to frequency. The elongated tails toward zero in (b) correspond to a trough or peak in (a). Some weak absorption lines that

are not visible in (a) are enhanced in visibility by the long tail in (b). The arrows indicate the selected online and offline laser wavelengths.'

Line 172-173: Please mark P(12) line on figure 1. What about other lines presented in figure, how do they compare to the selected P(10) line? Otherwise, limit the figure to the discussed lines.

This figure provides the reader with a comprehensive overview of the $CO_2$ absorption band, which we believe would be of interest to a general audience. Meanwhile, Fig. 3 delves into the specifics of the selected line. We reduced the wavelength span and marked each absorption lines.

**Section 2.5: Laser and Wavelength Locking**

Line 183-184: How the DAOD error, of less than $10^{-4}$, was evaluated?

This directly corresponds to Fig. 3b of the AOD sensitivity to the laser frequency. We added '(refer to Fig. 3b)' to make it clearer (line 184).

**Section 3.1: Error Analysis**

Line 193-198: Please define all symbols and discuss these equations.

Done

Line 199: Which results are referred to? Do you mean 'analysis'?

Changed 'these results' to 'Eq. (11b)'.

Line 206-207: Please include the error due to the initial pressure estimate used for the iterative process.

The initial pressure estimate is used to initiate the iterative procedure to derive the solution and does not introduce extra errors but may impact the number of iterations if it deviates significantly from the true value.

Line 209: Please include reference for <5% other pressure.

Done

**Section 3.2: Simulation Results**

Line 238: Please define the figure of merit (FOM).

We added to the sentence 'These advantages are summarized in Table 1' with 'and quantified by a figure of merit (FOM) in terms of SNR improvement.' (Line 255)

Line 240-241: Please state how the photon number per pulse was evaluated? Why was the photon number addressed here not signal as equation (1)?

Please refer to the new Appendix A for the derivation of the photon number. We have revised Eq. (1) to utilize photon numbers as the return signal for consistency with the subsequent equations.

Line 245: Typically, signal-induced shot noise is the dominant noise source for lidar systems, whereas daytime background can limit the dynamic range. Moreover, background blocking filters can resolve this issue as pointed out in (Line 294). Please investigate.

A lidar simulator, which includes those we have developed, typically considers all essential noise sources. The lidar community has extensively examined the influence of each noise source on lidar

measurements, and these effects are widely acknowledged within the community. The reviewer's comment on the dominance of signal shot noise primarily applies to robust lidar return signals, which is not a typical scenario for space lidar measurements.

Typically, the single-shot signal of lidar returns is highly noisy for space measurements due to the extended distance and the inverse proportionality to the squared range of the lidar return signal. Consequently, the consideration of detector noise, as well as solar background noise during daytime measurements, becomes crucial. Averaging over numerous shots and/or range bins becomes imperative, as demonstrated in this study (utilizing a horizontal average of 10 km for surface measurements and an additional vertical average of 1 km for atmospheric measurements). Signal-induced shot noise dominates only when the lidar return signal is strong, as observed in surface $CO_2$ and pressure measurements (Figs. 7e and 7f), where the lidar surface return is relatively robust. Consequently, nighttime and daytime measurements exhibit similar performance, as illustrated in Figs. 7d-f.

However, the return signal in the atmosphere is significantly weaker compared to the surface return, and signal shot noise is no longer the predominant factor. Consequently, the daytime SNR is smaller than the nighttime SNR in the atmosphere due to the undeniable impact of solar background noise, as depicted in Fig. 7b. Given that the primary focus of this paper is on the feasibility of DIAL measurements on Mars with the specified parameters in Table 2, we did not engage in a detailed discussion of the impact of each noise source on SNR in the main body of the original manuscript. However, to address this concern, we have introduced Appendix A, which includes additional discussions on the dependence of SNR on various noise sources.

The aforementioned information was briefly introduced in the discussion on simulation results in the original manuscript (e.g., lines 475-477, lines 291-292 in the original manuscript), and it should be apparent to those familiar with lidar detection.

Line 277: Please elaborate on solar background noise calculation and why it wasn't included in the error analysis presented in equations (9) to (12).

The solar background noise was implicitly included in the SNR term in Eq. (9) in the original manuscript and is now explicitly presented in the new Appendix A.

Line 289-291: Need a figure for simulating the lidar return signals and $CO_2$ DAOD and surface pressure retrievals to support these claims.

Lines 289-291 discuss the simulation results presented in Fig. 7. The simulation in Fig. 7 primarily centers on the impact of noise on the measurement to evaluate whether a desired precision is achievable with currently available parts and devices as listed in Table 2, where the telescope and laser are two key components that drive the cost of the system. Figs. 7b - 7d were calculated using Eq. (9) and Eq. (11) with SNR determined using Eq. (A1). We added two panels of simulated surface return signals and SNR to Fig. 7 and updated the caption accordingly.

**Figures and Tables**

Line 400: Figure 1: Please indicate the spectral resolution of the calculated cross section. It is unclear why wide spectral range is shown, rather than focusing on 1.9640146 μm line. This calculation focuses on the dominant spectral lines while ignoring weaker lines. Please include weaker line and plot in log scale to be comparable to the optical depth calculations presented in Figure 3.

Figure 1 encompasses a broad spectral range, while Fig. 3 narrows its focus to the selected $CO_2$ absorption line. All $CO_2$ absorption lines are included in both figures. Please refer to our responses to the general and specific comments above for further clarification.

Line 406: Figure 2: In addition, please include typical Martial vertical $CO_2$ profile as applied to equation (6).

We would like to clarify that Eq. (6) is used to calculate the weighted mean of Martian gravity acceleration, and it changes only slightly over time. Eq. 6 is not used in the SNR-based uncertainty estimation, which is the focus of this paper. On the other hand, the $CO_2$ number density is employed in calculating $CO_2$ AOD, and it can be readily derived from the pressure and temperature profile in Fig. 2 using the ideal gas law and $CO_2$ mixing ratio. To provide additional clarity, we have included a panel displaying the number density profile on Mars.

Line 408: Figure 3: How the absorption optical depth, presented in figure 3a, was calculated? Is it based on a model similar to equation (2)? Please state the altitude limits.

Added 'using Eq. (2) from 60 km as TOA' in the caption for Fig. 3.

Line 416: Figure 6: Please replace 'system' with systematic'. Check if these profiles calculated using HAPI or equations (11).

We confirm that they were calculated using HAPI.

Line 425: Table 2: Some of the parameters listed in this table were not discussed within the manuscript. Please include a discussion for how these parameters are relevant to the measurements. For example, the beam expander throughput, telescope diameter and clear area ratio, detector quantum efficiency and dark current. What is the meaning of the 'fill factor'? is the DRS APD a single detector or array? What is the detector noise-equivalent-power and how it influences the errors? Please define abbreviations.

Please also refer to the responses above regarding similar reviewer comments. In the simulation, a single element is utilized. The 'fill factor' quantifies the proportion of returning laser light effectively incident on the detector. The noise-equivalent-power (NEP) of the detector, as implied by its name, corresponds to the detector's noises which are inherently factored into the SNR calculation, as outlined in Appendix A. Further consideration of NEP is deemed unnecessary. Additionally, we have included the information that the HgCdTe APD detector is manufactured by Leonardo DRS Electro-Optical Infrared Systems (lines 280-281).